# Whole-genome resequencing reveals *Brassica napus* origin and genetic loci involved in its improvement

Kun Lu [1,2,3], Lijuan Wei[1,2], Xiaolong Li[4], Yuntong Wang[4], Jian Wu[5], Miao Liu[1], Chao Zhang[1], Zhiyou Chen[1], Zhongchun Xiao[1], Hongju Jian[1], Feng Cheng [5], Kai Zhang[1], Hai Du[1,2,3], Xinchao Cheng[3], Cunming Qu[1,2,3], Wei Qian[1,2,3], Liezhao Liu[1,2,3], Rui Wang[1,2,3], Qingyuan Zou[1], Jiamin Ying[1], Xingfu Xu[1,2], Jiaqing Mei[1,2], Ying Liang[1,2,3], You-Rong Chai[1,2,3], Zhanglin Tang[1,2,3], Huafang Wan[1], Yu Ni[1,2,3], Yajun He[1], Na Lin[1], Yonghai Fan[1], Wei Sun[1], Nan-Nan Li[2], Gang Zhou [4], Hongkun Zheng [4], Xiaowu Wang[5], Andrew H. Paterson[6] & Jiana Li[1,2,3]

*Brassica napus* ($2n = 4x = 38$, AACC) is an important allopolyploid crop derived from inter-specific crosses between *Brassica rapa* ($2n = 2x = 20$, AA) and *Brassica oleracea* ($2n = 2x = 18$, CC). However, no truly wild *B. napus* populations are known; its origin and improvement processes remain unclear. Here, we resequence 588 *B. napus* accessions. We uncover that the A subgenome may evolve from the ancestor of European turnip and the C subgenome may evolve from the common ancestor of kohlrabi, cauliflower, broccoli, and Chinese kale. Additionally, winter oilseed may be the original form of *B. napus*. Subgenome-specific selection of defense-response genes has contributed to environmental adaptation after for-mation of the species, whereas asymmetrical subgenomic selection has led to ecotype change. By integrating genome-wide association studies, selection signals, and transcriptome analyses, we identify genes associated with improved stress tolerance, oil content, seed quality, and ecotype improvement. They are candidates for further functional characterization and genetic improvement of *B. napus*.

[1] College of Agronomy and Biotechnology, Southwest University, Beibei, 400715 Chongqing, China. [2] Academy of Agricultural Sciences, Southwest University, Beibei, 400715 Chongqing, China. [3] State Cultivation Base of Crop Stress Biology for Southern Mountainous Land of Southwest University, Beibei, 400715 Chongqing, China. [4] Biomarker Technologies Corporation, 101300 Beijing, China. [5] Institute of Vegetables and Flowers, Chinese Academy of Agricultural Science, 100081 Beijing, China. [6] Plant Genome Mapping Laboratory, University of Georgia, Athens, Georgia 30605, USA. These authors contributed equally: Kun Lu, Lijuan Wei, Xiaolong Li. Correspondence and requests for materials should be addressed to X.W. (email: wangxw@mail.caas.net.cn) or to A.H.P. (email: paterson@uga.edu) or to J.L. (email: ljn1950@swu.edu.cn)

**B**rassica napus is an important oilseed crop, worldwide, and includes both tuberous (swede or rutabaga) and leafy (fodder rape and kale) forms used for animal fodder and human consumption[1]. During B. napus breeding, the undesirable oil components, erucic acid and aliphatic glucosinolate, in the seeds have undergone a dramatic reduction, whereas oil content, seed yield, and disease resistance have been significantly improved[2]. As described by the Triangle of U[3], three allopolyploid species, Brassica napus (AACC, $2n = 38$), Brassica juncea (AABB, $2n = 36$), and Brassica carinata (BBCC, $2n = 34$) each originated from hybridization between two of three ancestral diploid species, Brassica rapa (AA, $2n = 20$), Brassica oleracea (CC, $2n = 18$), and Brassica nigra (BB, $2n = 16$). The three Brassica diploid progenitors are themselves ancient polyploids that experienced a lineage-specific whole-genome triplication, making the allopolyploid descendant B. napus an excellent model for investigating processes of polyploid speciation, evolution, and selection.

As one of the earliest allopolyploid crops, B. napus was formed by hybridization of B. rapa and B. oleracea[1]. The estimated formation time of B. napus were ~6700[4] and ~7500 years ago[5] and 38,000−51,000 years ago[6], based on two synonymous substitution (Ks) estimations and a Bayesian Markov chain Monte Carlo (MCMC) simulation, respectively. The literatures recorded that winter B. napus was first cultivated in Europe[7]. Around the year 1700, spring B. napus was developed and spread to England in the late 18th century[8]. The semi-winter ecotype was mainly cultivated in China, which was introduced from Europe in the 1930–1940s[9]. However, no truly wild B. napus populations are known[7], leading to the precise identities of the two progenitors that hybridized to form B. napus remain elusive, as B. rapa and B. oleracea have morphologically diverse subspecies and have commonly been cultivated throughout Europe for hundreds of years. The natural hybridization among these species occurred occasionally under appropriate conditions. Although a recent study suggested that the B. napus A subgenome might be derived from the ancestor of European turnip (B. rapa ssp. rapa)[6], more evidence to support the conclusion need to be provided, due to only 5 B. napus and 27 B. rapa accessions were included in their analysis. Previous studies based on nuclear and chloroplast markers also suggested that B. napus may have developed from an interspecific cross between turnip and broccoli, or resulted from several independent hybridization events[10,11]. To further understand the evolution of B. napus, it is necessary to reveal whether wild species or domesticated donors were parental progenitors, which B. rapa and B. oleracea subspecies involved in the formation of B. napus.

Genome resequencing has been used to identify genes that have contributed to domestication and improvement in crop plants, such as rice (Oryza sativa)[12], tomato (Solanum lycopersicum)[13], and soybean (Glycine max)[14]. Reference genomes of the three Brassica progenitors and the allopolyploid species B. napus and B. juncea have been released[4–6,15–17]. Resequencing of 199 B. rapa and 119 B. oleracea accessions[18] will facilitate investigations of selection acting in each B. napus subgenome and will clarify the progenitors of the species.

Here, we perform genome sequencing of 588 B. napus accessions and transcriptome sequencing of 11 tissues from two B. napus accessions having different seed quality. The large number of variations identified not only provide insight into the origin and evolutionary history of B. napus, identifying genetic loci involved in its improvement, but also lay a foundation for further functional validation of candidate genes controlling important traits. Furthermore, the findings of this study provide a basis to facilitate breeding of B. napus and related crops with favorable traits.

## Results

**Sequencing and variation discovery**. We resequenced 588 diverse B. napus accessions from 21 countries (Fig. 1a; Supplementary Tables 1; Supplementary Data 1) and obtained 4.03 Tb of clean data. After filtering, we aligned reads to the B. napus reference genome[5]. The mapping rate varied from 79.84 to 99.45%, and the effective mapped read depth averaged ~5× and ranged from 3.37× to 7.71×. We generated 5,294,158 single-nucleotide polymorphisms (SNPs; denoted as Bna) and 1,307,151 indels (Supplementary Data 2 and 3). Validation of 103 randomly selected SNPs in 20 accessions by Sanger sequencing indicated that most of the identified SNPs (95.1%) were authentic (Supplementary Data 4). The reliability of SNPs was further confirmed in that most SNPs (93.5 to 96.4%) were repeated in biological replicates of 20 accessions.

We aligned the B. napus data to a B. napus ancestral pseudo-genome (merging the B. rapa and B. oleracea reference genomes; Supplementary Fig. 1) and divided the SNPs into BnaA and BnaC two sets, based on the two progenitors. B. rapa and B. oleracea sequencing data were mapped onto the corresponding reference genomes. Then, the SNPs called from B. rapa were combined with BnaA to form the A (B. rapa and B. napus) subgenome SNP set (denoted as BraA, including 529,771 SNPs). Similarly, the C (B. oleracea and B. napus) subgenome SNP set (denoted as BolC), including 675,457 SNPs, was obtained (Supplementary Fig. 1).

**Origin of B. napus**. Based on the optimal models (GTR + F + ASC + R5, GTR + F + ASC + R7, and TVM + F + ASC + R8 for the Bna, BraA, and BolC SNP sets, respectively) (Supplementary Data 5–7), we constructed maximum likelihood (ML) trees, using IQ-TREE[19] (Figs 1a, 2a and 3a), resulting in divergent clades of B. napus and B. rapa or B. oleracea, respectively. Most B. napus accessions clustered together, based on ecotype, whereas clustering of B. rapa and B. oleracea largely reflected subspecies relationships. From the BraA phylogenetic tree, the B. napus clade was closest to the European turnip and distant from B. rapa ssp. rapa (Asian turnip) and other subspecies (Fig. 2a), consistent with a recent study suggesting that the B. napus A subgenome evolved from the ancestor of European turnip[6].

Demographic modeling, using ∂a∂i[20] and fastsimcoal2[21], supported the aforementioned results, and the log-likelihood values of the two optimal models were −11,826 and −4,194,976 (model a in Supplementary Fig. 2, model e in Supplementary Fig. 3). The best model in fastsimcoal2 also suggested that a gene flow event from European turnip to the B. napus A subgenome occurred ~106–1170 years ago (Supplementary Fig. 3).

Principal component analysis (PCA; Fig. 2b) located B. napus landraces ($n = 50$, registration date before 1980) near the European turnip ($n = 33$; Fig. 2b). Population structure analysis revealed that two population clusters ($K = 2$) represent the optimal model (Fig. 2c, Supplementary Fig. 4), which clearly separate B. napus landraces and European turnip from other B. rapa subspecies, supporting the evolutionary history of the B. napus A subgenome[6].

In the BolC ML tree, the B. napus accessions were closest to a B. oleracea branch comprising four subspecies-kohlrabi (B. oleracea var. gongylodes), broccoli (B. oleracea var. italica), cauliflower (B. oleracea var. botrytis), and Chinese kale (B. oleracea var. alboglabra)–suggesting that the B. napus C subgenome might have evolved from the common ancestor of these lineages (Fig. 3a). To validate this result, we performed demographic analyses using ∂a∂i[20] and fastsimcoal2[21] (Supplementary Figs. 3, 5). The ∂a∂i results supported the model that the B. napus C subgenome evolved from the common ancestor of kohlrabi, broccoli, and cauliflower (log-likelihood=−6806). In

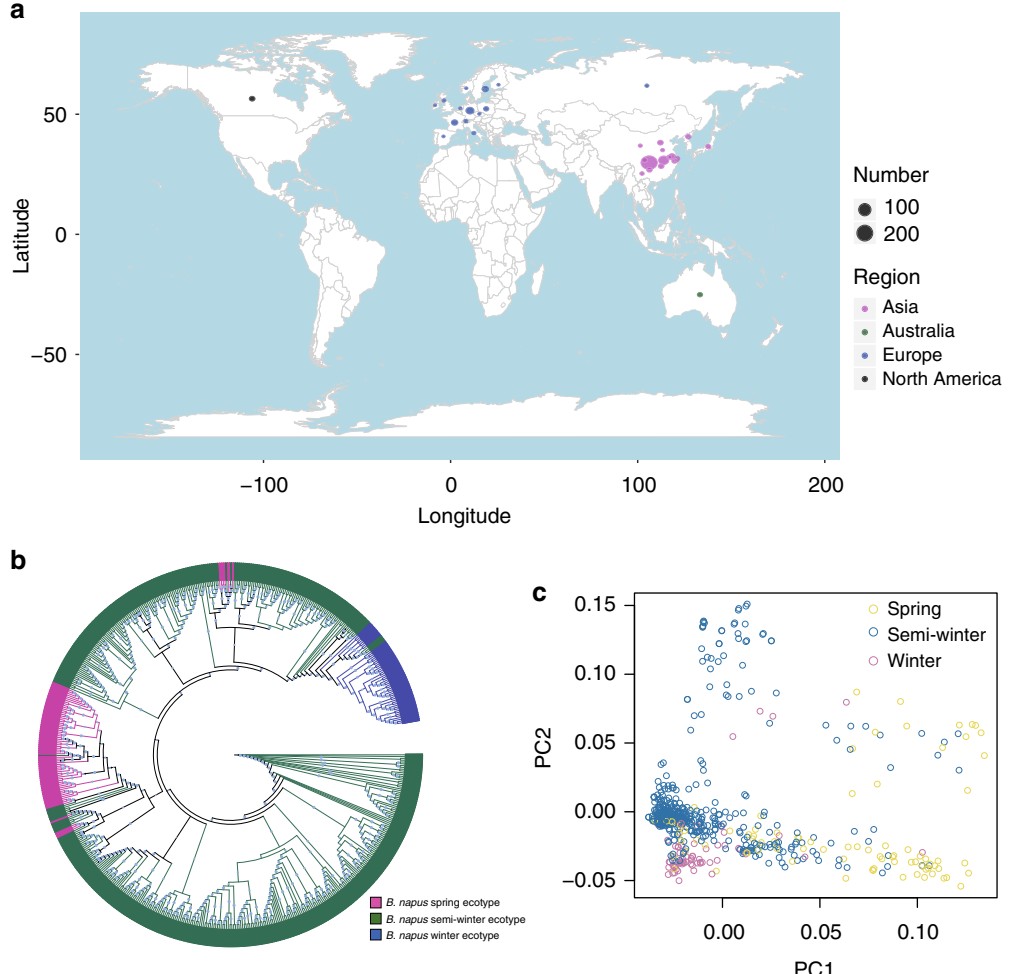

**Fig. 1** Geographic distribution and population structure of *B. napus* accessions. **a** Geographic distribution of 588 *B. napus* accessions in the world. **b** ML tree of all *B. napus* accessions inferred from SNPs at fourfold-degenerate sites. Phylogenetic tree was constructed using IQ-TREE[19] with the best model GTR + F + ASC + R5. Clades with bootstrap values of above 50% are indicated by a circled blue dot. **c** PCA plot of all the *B. napus* accessions used in this study. **a** was generated in R (V3.2.5) using package rworldmap. Source data are provided as a Source Data file

the fastsimcoal2 results, model a, which suggests that the *B. napus* C subgenome originated from the common ancestor of kohlrabi, cauliflower, and broccoli, was supported with a much higher log-likelihood ($\Delta = 460{,}3200$; log-likelihood=$-2{,}975{,}039$) than model b, suggesting that the *B. napus* C subgenome originated during divergence of ancestors of four *B. oleracea* subspecies.

Considering that migration events may have occurred at three different stages (recent, medium- and long-term), five models (models c–g) were compared to infer the evolutionary history of the *B. napus* C subgenome. Our data support model c (log-likelihood=$-2{,}914{,}388$), indicating that the ancestor of *B. napus* split from the common ancestor of four *B. oleracea* subspecies, with recent gene flow into *B. napus* ~108–898 years ago (Supplementary Fig. 6).

In the PCA plot, kohlrabi ($n = 19$) was the closest subspecies to *B. napus* landraces, followed by Chinese kale, broccoli, and cauliflower (Fig. 3b), suggesting that *B. napus* landraces were closer to the four *B. oleracea* subspecies. The optimal number of population clusters of *B. napus* landraces and *B. oleracea* was identified as $K = 2$ (Fig. 3c; Supplementary Fig. 4b), representing distinct *B. napus* and *B. oleracea* clusters, indicating that the *B. napus* C subgenome has a more complex origin than A subgenome.

The *Cruydt Boeck* recorded the early form of winter *B. napus* and noted that its raepolie was used as lamp fuel[7]. To validate this

record, we compared the linkage disequilibrium (LD) decay of different groups, based on progenitors of the A and C subgenomes (33 accessions of European turnip and 66 accessions of four *B. oleracea* subspecies) and *B. napus* accessions in the three SNP sets (Supplementary Table 2). At a threshold of $r^2 = 0.3$, the LD decay in *B. rapa* (2.10 kb) and *B. oleracea* (27.90 kb) was stronger than those in the A (19.30 kb) and C (1365.30 kb) subgenomes of *B. napus*, respectively (Supplementary Table 2; Supplementary Figs. 7–9), supporting a strong bottleneck developed in both two subgenomes during *B. napus* evolution. In the Bna SNP set, the oilseed *B. napus* showed stronger LD decays than did fodder and vegetable accessions, and the LD decays of winter type were also stronger than those of the spring and semi-winter types (Supplementary Table 2).

We estimated the historical effective population sizes ($Ne$) and divergence times for different ecotypes and usages of *B. napus* using SMC++[22]. The $Ne$ values for different *B. napus* ecotypes showed similar dynamics (Supplementary Fig. 10). The estimated time of *B. napus* formation was ~1912–7178 years ago. The winter and semi-winter *B. napus* ecotypes diverged ~60 years ago, whereas the winter and spring *B. napus* diverged ~416 years ago, and oilseed and non-oilseed *B. napus* diverged ~277 years ago. These results are consistent with historical records[8,9]. Based on the aforementioned results, we could speculate that winter oilseed was the original *B. napus*.

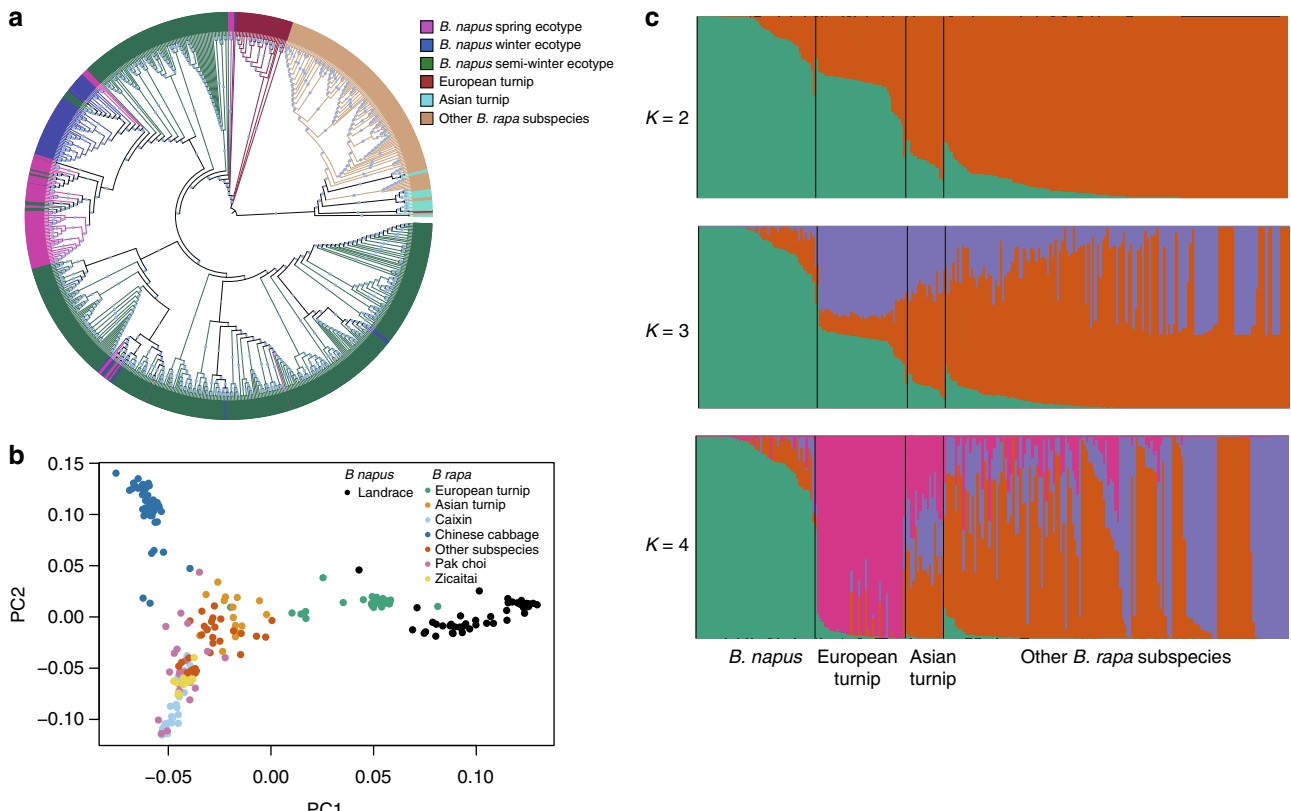

**Fig. 2** Population structure of 588 *B. napus* accessions and 199 of *B. rapa* accessions. **a** ML tree of all *B. napus* accessions and 199 *B. rapa* accessions inferred from SNPs at fourfold-degenerate sites. Phylogenetic tree was constructed using IQ-TREE with the best model GTR + F + ASC + R7. Clades with bootstrap values of above 50% are indicated by a circled blue dot. **b** PCA plot of 50 *B. napus* landraces and 199 *B. rapa* accessions. **c** Model-based Bayesian clustering of 50 *B. napus* landraces and 199 *B. rapa* accessions performed using STRUCTURE 2.1 with the number of ancestry kinships (*K*) set to 2, 3, or 4. Each accession is denoted by a vertical bar composed of different colors in proportions corresponding to its proportion of genetic ancestry from each of these ancestral populations. Source data are provided as a Source Data file

**Selection signals under improvement of *B. napus*.** We divided the *B. napus* improvement process into two stages after *B. napus* origin. The first stage of improvement (FSI) was the process from original *B. napus* to landrace, while the second stage of improvement (SSI) represented the improvement from landraces to improved cultivars. As no wild *B. napus* populations are known[7], we pooled two groups of progenitors, European turnip (*n* = 33) and four *B. oleracea* subspecies (*n* = 66), to represent the pseudo-wild ancestral A (denoted AA) and C (denoted CA) subgenomes of *B. napus*, respectively. To avoid the influence of introgression derived from non-ancestors of progenitors, we identified FSI-selection signals from the shared variations between *B. napus* landraces and their ancestors by comparing genomic variations with high fixation index ($F_{ST}$) and reduction of diversity (ROD) between the A subgenome of *B. napus* landraces (denoted AL) and AA, and between the C subgenome of *B. napus* landraces (denoted CL) and CA, using *B. napus* landraces as the derived group and progenitors as the control group. SSI-selection signals were detected by comparing *B. napus* landraces with improved cultivars (*n* = 95, registration date after 2000) and by comparing cultivars with different ecotypes.

Nucleotide diversity (π) decreased from $7.23 \times 10^{-4}$ in AA to $5.40 \times 10^{-4}$ in AL and from $7.45 \times 10^{-4}$ in CA to $4.97 \times 10^{-4}$ in CL (Supplementary Data 8 and 9), implying that during the FSI more genetic diversity was lost in the *B. napus* C subgenome than in the A subgenome. The $F_{ST(AL/AA)}$ was 0.136 (Supplementary Data 10), lower than $F_{ST(CL/CA)}$ (0.246) (Supplementary Data 11). Notably, both $F_{ST(AL/AA)}$ and $F_{ST(CL/CA)}$ were higher than $F_{ST(improved\ cultivar/landrace)}$ (0.016) (Supplementary Fig. 11;

Supplementary Data 12), suggesting that less genetic differentiation occurred during the SSI than during the FSI. This difference is most likely attributable to the most noteworthy event in the breeding history of *B. napus*: namely the widespread use of the erucic-acid-free variety Liho and the low-glucosinolate variety Bronowski to breed cultivars with zero erucic acid and low glucosinolate[23].

Calculation of the *z* transformations of $F_{ST}$ and ROD detected 424 and 366 common FSI-selection windows, corresponding to 66 and 44 selection signals in the A and C subgenomes, respectively (Supplementary Figs. 12, 13; Supplementary Data 13 and 14). In the outlier regions, we retrieved 1522 genes in AL and 811 genes in CL (Supplementary Data 15 and 16), respectively.

Gene ontology (GO) enrichment analysis revealed that genes in the $F_{ST}$ and ROD overlapping outliers of the A subgenome were enriched in three groups of GO classifications, including stress tolerance (response to herbivore (GO:0080027), abscisic acid metabolic process (GO:0009687), and regulation of immune response (GO:0050776)), development (axillary shoot meristem initiation (GO:0090506) and meristem initiation (GO:0010014)), and metabolic pathways (unsaturated fatty acid biosynthetic process (GO:0006636), triglyceride biosynthetic process (GO:0019432) and terpenoid metabolic process(GO:0006721)) (Supplementary Data 17), whereas those of the C subgenome were over-represented in development process, photomorphogenesis (GO:0009640), trichome morphogenesis (GO:0010090), regulation of cell proliferation (GO:0042127), and inflorescence development (GO:0010229; Supplementary Data 18). These results suggest that A subgenome-specific selection may have

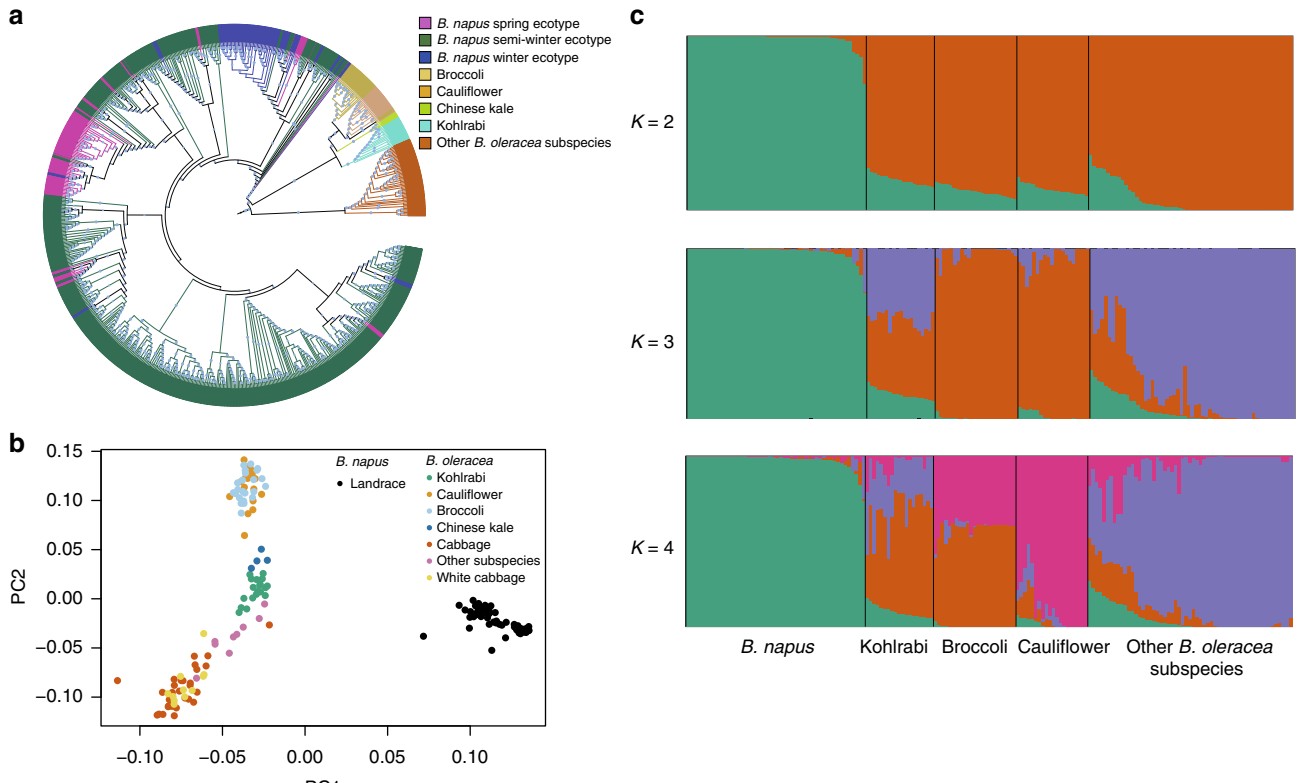

**Fig. 3** Population structure of 588 *B. napus* accessions and 119 of *B. oleracea* accessions. **a** ML tree of all *B. napus* accessions and 119 *B. oleracea* accessions inferred from SNPs at fourfold-degenerate sites. Phylogenetic tree was constructed using IQ-TREE with the best model TVM + F + ASC + R8. Clades with bootstrap values of above 50% are indicated by a circled blue dot. **b** PCA plot of 50 *B. napus* landraces and 119 *B. oleracea* accessions. **c** Model-based Bayesian clustering of 50 *B. napus* landraces and 119 *B. oleracea* accessions performed using STRUCTURE 2.1 with the number of ancestry kinships (K) set to 2, 3, or 4. Each accession is denoted by a vertical bar composed of different colors in proportions corresponding to its proportion of genetic ancestry from each of these ancestral populations. Source data are provided as a Source Data file

promoted the stress tolerance and oil accumulation during the FSI of *B. napus*, whereas improvement of its developmental traits may have been due to asymmetrical selection in the C subgenomes.

Regarding SSI-selection signals, we identified 912 outliers (corresponding to 79 selection signals spanning 16.35 Mb) simultaneously detected by at least three of the four comparisons ($F_{ST}$, ROD, XP-CLR, and XP-EHH) (Fig. 4a–d; Supplementary Fig. 14; Supplementary Data 19). The 2610 genes identified in the overlapping regions were over-represented in phloem glucosinolate transport (GO:1901349) and glucosinolate biosynthetic process (GO:0019761) — representing key genes whose silencing might have contributed to the reduced seed glucosinolate content of elite *B. napus* (Supplementary Data 20 and 21). These genes were also enriched in sucrose transport (GO:0015770), RNA splicing (GO:0008380), and pollination (GO:0009856), indicating that not only seed quality but also fertility, energy utilization efficiency, and post-transcriptional regulation were improved in the breeding programs. In addition, 84 selection signals between double-high (high erucic acid and glucosinolate) and double-low (low erucic acid and glucosinolate) cultivars were merged from 1417 overlapping outlier windows (Supplementary Fig. 15; Supplementary Data 22 and 23), and had similar GO enrichment to that in the SSI. We determined that the reproductive system of *B. napus* may have been optimized, as genes in the outliers were also enriched in the reproductive process (GO:0022414), reproductive structure development (GO:0048608) and regulation of secondary shoot formation (GO:2000032) (Supplementary Data 24).

Furthermore, candidate genes involved in glucosinolate transport and fatty acid elongation were located within syntenic regions in the A and C subgenomes (Supplementary Data 20 and 23), implying that parallel subgenomic selection may have contributed to improved seed quality.

**Genome-wide association studies.** To identify candidate genes responsible for 11 important traits, we conducted multi-locus random mixed linear model analysis for genome-wide association studies (GWAS) using mrMLM v1.3[24]. To obtain high-quality SNPs, we performed imputation for the Bna SNP set, and retained 670,028 SNPs with a MAF of >5% for GWAS. Comparison between imputation results for 19 polymorphic nucleotides and Sanger sequencing results indicated that 98.74% of imputed genotypes were correct (Supplementary Data 25). The average correlations ($r^2$) between imputed and true genotypes for two biological replicates of 20 accessions was 0.956 with minimum and maximum values ranging from 0.928 to 0.967 (Supplementary Fig. 16), further confirming the accuracy of the imputed genotypes. Using the Genetic Type I Error Calculator (GEC)[25], the effective number of independent SNPs was 206,001, and the P-value thresholds used to identify significant loci were set at $2.43 \times 10^{-7}$ [significant, $0.05/n$, $-\log10(P) = 6.61$] and $4.85 \times 10^{-6}$ [suggestive, $1/n$, $-\log10(P) = 5.31$].

In total, we identified 60 loci significantly associated with 10 target traits, including 5 related to seed yield, 3 to silique length, 4 to oil content, and 48 to seed quality (Fig. 4c–f; Fig. 5b; Supplementary Fig. 17; Supplementary Data 26). In addition, three loci were shown to be associated with flowering time at the

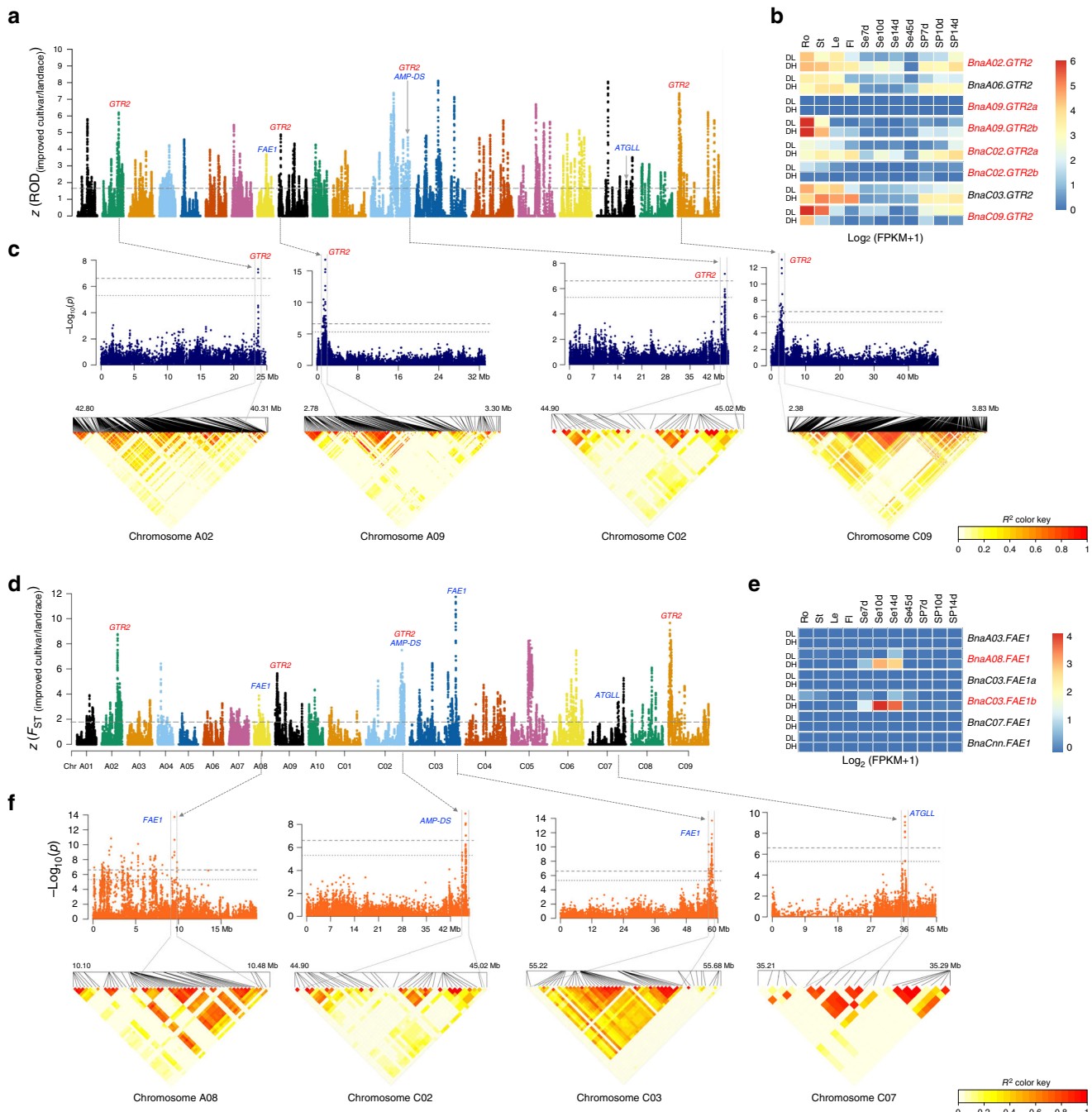

**Fig. 4** Genome-wide scanning and annotations of selected regions during the SSI of *B. napus*. **a**, **d** Genome-wide screening of SSI-selection signals with ROD and $F_{ST}$. ROD and $F_{ST}$ were normalized as *z* scores for *B. napus*. A 100-kb sliding window with an increment of 10 kb was used to calculate these values. Each point represents a value in a 100-kb window. Horizontal dashed lines show the significance level of $\alpha = 0.05$, corresponding to $z = 1.645$. The glucosinolate transport genes in the selection outlier regions are labeled in red, and erucic acid biosynthesis genes are in blue. **b**, **e** Expression patterns of two *GTR2* and *FAE1* family members. Genes in the selection outlier regions during the SSI are in red. DH double-high accession Zhongyou821, DL double-low accession Zhongshuang11, Ro root, St stem, Le leaf, Fl flower, Se7d, 10d, 14d, and 45d seeds at 7, 10, 14, and 45 days after flowering, SP7d, 10d, and 14d silique pericarp at 7, 10, and 14 days after flowering. **c**, **f** GWAS results of total glucosinolate and erucic acid content that overlapped selection signals. Dashed horizontal lines depict the significant ($-\log 10(P) = 6.61$) and suggestive ($-\log 10(P) = 5.31$) thresholds. The lower panel shows the LD blocks of significantly associated loci in GWAS. Source data are provided as a Source Data file

suggestive threshold. SNPs significantly associated with seed quality traits explained 14.30–35.47% of the phenotypic variance, higher than those for seed yield (16.81%) and oil content (20.64%). Most loci that were associated with seed quality in a previous GWAS[26] were also detected here. As traits related to oil content and yield are complex and easily affected by

environmental variation, we integrated the significant loci from this study with quantitative trait loci (QTLs) identified previously[27,28]. Candidate genes controlling different traits thus were identified based on functional annotation and transcriptome analysis in intersecting regions of selection outliers and QTL, as discussed below.

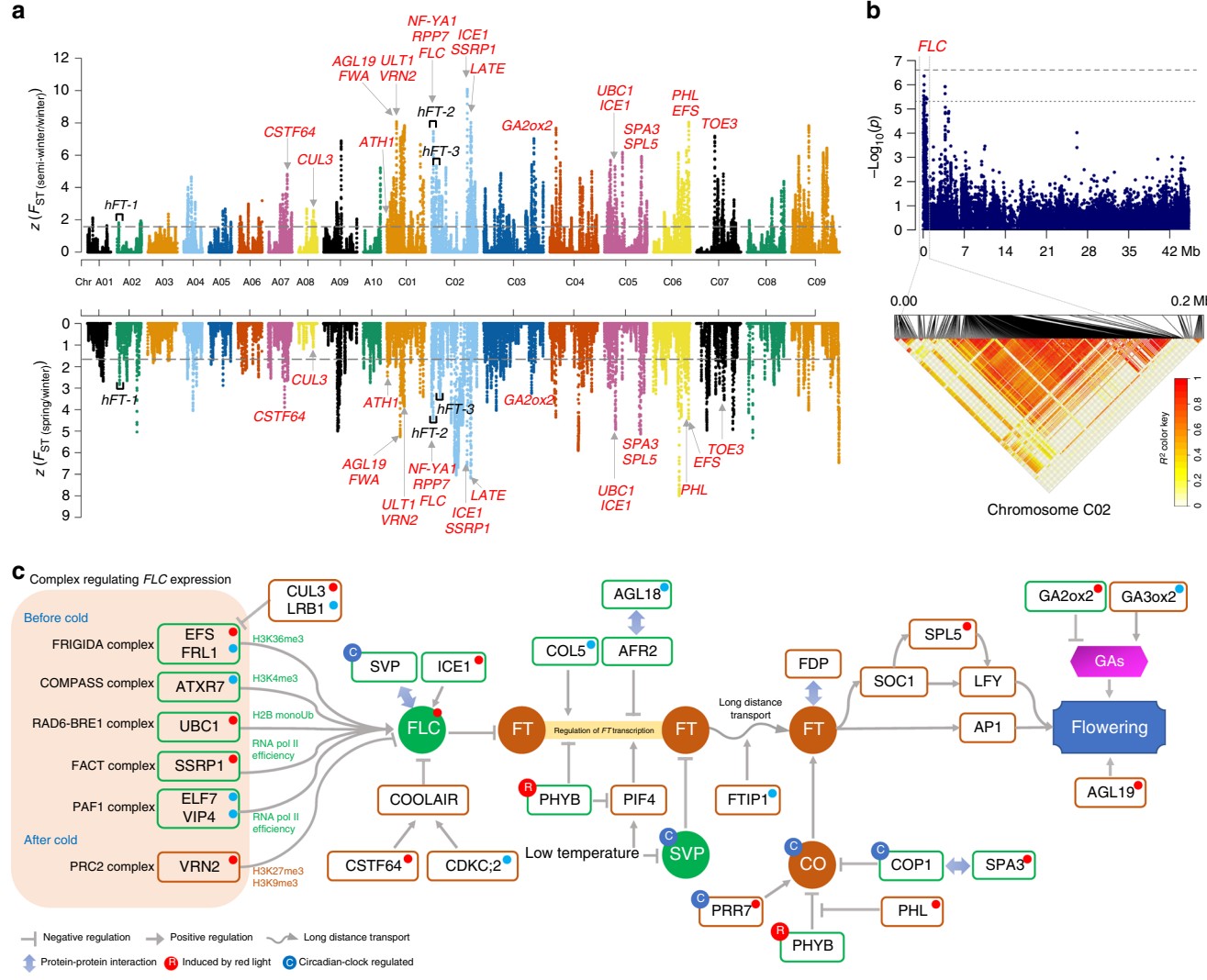

**Fig. 5** Overview of flowering-time regulation under the selection of the ecotype improvement of *B. napus*. **a** Genome-wide screening of ecotype improvement selection signals of $F_{ST\ (semi-winter/winter)}$ and $F_{ST\ (spring/winter)}$. Flowering-time genes simultaneously identified in selection outlier regions between winter and spring ecotypes, and between winter and semi-winter ecotypes are labeled in red in the corresponding chromosomes. Full descriptions of these genes are shown in Supplementary Data 36. Three LD blocks containing significant GWAS associations for flowering time are labeled in black. **b** GWAS results of flowering time that overlapped with selection signals. Dashed horizontal lines depict the significant ($-\log10(P) = 6.61$) and suggestive ($-\log10(P) = 5.31$) thresholds. Lower panel shows the LD blocks of significantly associated loci in GWAS. **c** Major flowering-time pathways and selective effects acting on their components during ecotype improvement. Brown and green rounded rectangles represent proteins involved in positive and negative regulation of flowering time, respectively. Flowering integrator proteins are shown in brown and green dots, respectively. Red dots at the top right of gene symbols represent flowering-time-regulation genes simultaneously identified in the selection outlier regions both semi-winter/winter and spring/winter comparisons, and purple and blue dots represent those of spring/winter and semi-winter/winter comparisons, respectively. Source data are provided as a Source Data file

**Transcriptome analysis**. To identify differentially expressed SSI-related genes, we harvested 11 tissues from a high-oil-content and double-low cultivar Zhongshuang11 and a low-oil-content and double-high cultivar Zhongyou821. After filtering, 848,776,815 125-bp paired-end reads were obtained, containing 212.19 Gb of data. On average, 82.64% of the reads uniquely mapped to the *B. napus* reference genome (Supplementary Data 27)[5]. The highest number of differentially expressed genes (DEGs) (Supplementary Fig. 18) was detected in seeds at 45 days after flowering (Se45D), with 9962 up- and 12,441 down-regulated genes. The lowest number of DEGs occurred in Se14D, when only 2420 genes were differentially expressed.

Genes that were down-regulated in the stem (St), leaf (Le), and silique pericarp at 7 (SP7D), 10 (SP7D), and 14 (SP7D) days after flowering in Zhongshuang11 were significantly over-represented

in the glucosinolate biosynthetic process (GO:0019761; Supplementary Figs. 19–29; Supplementary Data 28), indicating that the lower seed glucosinolate content in the double-low accession was likely caused by reduced glucosinolate biosynthesis. In silique pericarp, up-regulated genes were significantly over-enriched in the fatty acid biosynthetic process, suggesting that the silique pericarp may also play an important role in increasing seed oil content.

**Major genes improving environmental adaption**. During the FSI, defense responsive genes may play key roles in *B. napus* environmental adaptation. In the A subgenome, we identified 6, 5, 6, and 14 genes associated with drought tolerance, herbivore resistance, responses to mechanical stimulus, and immune responses, respectively (Supplementary Data 29). Among the

genes related to drought tolerance, *NCED3* (*Bra001552*) plays a major role in the regulation of ABA (abscisic acid) biosynthesis in response to water deficit[29]. Besides, *NCED5* (*Bra032359*) contributes, along with *NCED3*, to ABA production, thereby affecting plant growth and water stress tolerance[30].

Endogenous ABA is rapidly produced during drought, inducing stomatal closure and thereby enhancing drought tolerance. A subgenome-specific selection on ABA biosynthesis pathway genes might have been important for enhancing drought tolerance during the FSI of *B. napus*, and laid a solid foundation for its cultivation in diverse environments. Other candidate genes involved in ABA metabolism, are also noteworthy, such as *CYP707A3* (*Bra021965* and *Bra025083*), *XERICO* (*Bra013211*), and *PHYB* (*Bra001650*), all of which are associated with drought stress responses, via regulation of ABA accumulation (Supplementary Data 29).

In addition, A subgenome-specific selection also contributed to disease resistance improvement during the FSI of *B. napus*. In the selection regions, *NPR3* (*Bra025093*) is one of the notable genes. As a receptor for the immune signal salicylic acid in plants, NPR3 controls the proteasome-mediated degradation of NPR1, which is involved in negative regulation of defense responses against bacterial and oomycete pathogens[31]. Similar defense-responsive genes, *BAH1*(*Bra032581*), *NHL25* (*Bra028103*), and *NDR1* (*Bra035766*) are also involved in regulating plant innate immunity to microbes (Supplementary Data 29) and may have contributed to biotic stress during the FSI of *B. napus*.

**Major genes improving seed quality**. In *B. napus*, seed oil contains 50% erucic acid and high levels of glucosinolates that significantly reduce nutritional value. Selection signatures and GWAS results (Fig. 4c–f) implicate several candidate genes associated with seed erucic acid and glucosinolate content.

Glucosinolates are synthesized in rosette leaves and silique walls and subsequently relocated to the embryo through the phloem by transporters. We identified the glucosinolate-specific transporter, GTR2, as likely responsible for glucosinolate delivery from source tissues into seeds in *B. napus*[32]. Among the eight *Bna.GTR2* genes, three (*BnaC02g42280D*, *BnaA09g06180D*, and *BnaA09g06190D*) are silenced in the seeds of the double-low cultivar, and the other five are moderately expressed (Fig. 4b); of these, three (*BnaA02g33530D*, *BnaC02g42260D*, and *BnaC09g05810D*) are located in major QTL regions. Further reduction of anti-nutritional glucosinolates, in *B. napus* seeds, may be achieved through silencing these five *GTR2* genes, especially those located in major QTL regions.

Four GWAS signals associated with erucic acid were identified during the SSI, including two on chromosomes A08 (*BnaA08g11130D*) and C03 (*BnaC03g65980D*) (Fig. 4) near the active *FATTY ACID ELONGASE 1* (*FAE1*) genes whose products control erucic acid synthesis from oleoyl-CoA (C18:1-CoA)[33]. These two genes were significantly highly expressed in the high-erucic acid cultivar, suggesting that *FAE1* silencing is an important contributor to reduced seed erucic acid content. Additional GWAS signals associated with erucic acid were on chromosome C02 and C07, (Fig. 4), near *AMP-DS* (*BnaC02g42510D*) and *ATGLL* (*BnaC07g30920D*) and may represent SSI-selection genes whose products reduce seed erucic acid.

**Genes involved in ecotype improvement of *B. napus***. Based on vernalization requirement, *B. napus* is divided into winter (requiring a prolonged cold period), semi-winter (requiring a short cold period), and spring (no cold requirement) ecotypes. The original *B. napus* was a winter oilseed; spring and semi-

winter ecotypes were developed for adaptation to different environments. To identify candidate genes responsible for ecotype improvement, we scanned the selection outliers using the winter ecotype as the control and the spring and semi-winter ecotypes as derived groups (Fig. 5a; Supplementary Figs. 30−31). Comparisons between *B. napus* winter and semi-winter, and between winter and spring ecotypes detected 1996 and 1117 overlapping outlier windows, including 4548 genes in 156 selection regions and 2729 genes in 107 selection regions, respectively (Supplementary Data 30−33). The majority of selection regions were located in the C subgenome, and only 32 and 21 were distributed on the A subgenome, respectively, suggesting that ecotype improvement was mainly caused by asymmetrical subgenomic selection. Besides, 844 outlier windows corresponding to 72 selection regions were overlapped between two ecotype improvement analyses, these parallel selection signals might contribute to local adaption of *B. napus*.

Genes in the ecotype improvement selection regions were over-represented in maintenance of floral organ identity (GO:0048497), floral organ abscission (GO:0010227), and regulation of floral meristem growth (GO:0010080), suggesting that these SSI-selection signals might be critical for local environmental adaptation of *B. napus* (Supplementary Data 34 and 35).

We retrieved 306 genes related to flowering time in *A. thaliana* from FLOR-ID (FLOweRing Interactive Database)[34] and identified 1225 orthologs in *B. napus*. Among the 24 flowering-time genes simultaneously located within outlier regions of the two comparisons (Fig. 5a–c), 22 of which were distributed on the C subgenome and regulating flowering time mainly through vernalization and photoperiod pathways (Supplementary Data 36). In particular, the *FLC* locus and its chromatin-mediated regulators will be the most promising targets for regulating flowering time of *B. napus*.

In the vernalization pathway, *FLC* (*BnaC02g00490D*) encodes a core regulator in *B. napus* ecotype improvement (Fig. 5c), as the product of the homologous *Arabidopsis thaliana FLOWERING LOCUS C* (*FLC*) is an important repressor of floral transition[35]. A recent study reported that different *FLC* paralogs contribute differentially to natural variation in *B. napus* flowering time, and that a 2.833-kb insertion in *BnFLC.A2* and its homeologous exchange (HE) with *BnFLC.C2* generated early-flowering *B. napus* ecotpyes[36]. Our GWAS detected three loci associated with flowering time at the suggestive threshold (Fig. 5b). Interestingly, two *FLC* genes on chromosome A02 and C02, located within the two LD blocks (*hFT-1* and *hFT-2*) controlling flowering time variation, each explained ~8% of the total phenotypic variance (Supplementary Data 27). These results further confirmed the importance of *FLC* during ecotype improvement.

Genes involved in chromatin modification also experienced extensive selection for ecotype improvement, via regulation of *FLC* expression (Fig. 5a–c), such as *FLC* activator gene *SSRP1* (*BnaC02g37430D*), encoding components of the facilitates chromatin transcription (FACT) complex, and *FLC* repressor gene *VRN2* (*BnaC01g21540D*), encoding a component of the polycomb repressive complex 2 (PRC2).

In *Arabidopsis*, the histone chaperones SSRP1 in the FACT complex assist the progression of RNA polymerase II and promote *FLC* expression[37]. In contrast, VRN2 plays a critical role in maintaining transcriptional repression of *FLC*, after vernalization, by regulating the deposition of H3K27me3 and H3K9me3 repressive marks at the *FLC* locus[38]. These results suggest that epigenetic variations, especially histone modifications involved in regulating *FLC* expression, are closely associated with flowering time in *B. napus*. These genes therefore represent promising regulatory targets in future *B. napus* breeding programs (Fig. 5c; Supplementary Data 36).

**Major genes for seed oil content and seed yield**. Seed oil content may have been greatly improved during the FSI of *B. napus*. We detected 9 FSI selection signals that overlapped with oil content QTLs (Supplementary Data 37). In these overlapping regions, two genes were involved in fatty acid synthesis, one in triacylglycerol biosynthesis, and seven in fatty acid elongation. For example, (*KASI*) on chromosome A02 (Supplementary Data 20 and 23) encodes a protein that may be crucial for fatty acid synthesis and also plays a role in chloroplast division and embryo development[39].

Mutation of *Arabidopsis KASI* disrupted embryo development and dramatically reduced fatty acid levels in seeds. *Bna.FATA1* (BnaA03g37700D) is an ortholog of *Arabidopsis FATA ACYL-ACP THIOESTERASE 1* on chromosome A03 (Supplementary Data 20 and 23) whose product controls termination of intraplastidial fatty acid synthesis by hydrolyzing the acyl-ACP complexes. Mutation of two *Arabidopsis FATA* genes reduces FATA activity, affecting oil content and seed fatty acid composition[40].

*B. napus* seed yield is influenced by silique length and shoot branching. We detected three loci significantly associated with silique length. The peak of significant SNP-trait associations was located at 27.99 Mb on chromosome A09, near a 165-bp deletion in *ARF18* that is associated with increased seed weight and silique length, via maternal regulation[41]. We also identified two pairs of parallel SSI-selection signals in double-low *B. napus* cultivars. The orthologs of *Arabidopsis BRC1*[42] on chromosomes A01 and C01 and *DOF4.4*[43] on chromosomes A08 and C03 (Supplementary Data 23) may encode key regulators of shoot branching and silique development in outlier regions. Hence, manipulating genes associated with seed oil content and seed yield within the SSI-selection signals and of candidate genes identified by GWAS should have practical applications in *B. napus* breeding for high yield and oil content.

## Discussion

In this study, we developed a large genome variation data set for genetically diverse *B. napus* accessions, which provided an opportunity to finely resolve the origin and evolutionary history of *B. napus*. Based on aforementioned analyses, we posit that the *B. napus* was originated from the hybridization between domesticated *B. rapa* and *B. oleracea* ~1910–7180 years ago (Supplementary Fig. 32), which accorded with previous conclusions (~6700 and 7500 years ago) derived from Ks estimation[4,5]. The *B. napus* A subgenome evolved from the ancestor of European turnip; and the *B. napus* C subgenome might have evolved from the common ancestor of kohlrabi, cauliflower, broccoli, and Chinese kale. In addition, the LD and demographic analyses support that the original *B. napus* was winter oilseed, and the spring and semi-winter *B. napus* developed ~416 and ~60 years ago, and non-oilseed *B. napus* developed ~277 years ago (Supplementary Fig. 32). These results are consistent with historical records, which indicate that spring *B. napus* developed around the year 1700[8], and that the semi-winter ecotype has only a short history in China, and arose from the winter ecotype, which was introduced from Europe in the 1930–1940s[9]. In recent 1000 years, gene flow from two progenitors into *B. napus* also occurred occasionally, leading to improvement of complex traits in *B. napus*.

Previous archeological and linguistic lines of evidence suggest that turnip is likely the first domesticated *B. rapa* in the European-Central Asian region[44,45]. A recent demographic inference further supported an eastward series of *B. rapa* domestication events, over the past several thousand years, and rapini (*B. rapa* ssp. *sylvestris*), which split from the European-

Central Asian *B. rapa* (European turnip) cluster, ~3715–6190 years ago, is not likely a wild *B. rapa*[46]. Since the estimated origin times of *B. napus* in previous and our studies were all earlier than the divergence time between rapini and the European-Central Asian *B. rapa*, European turnip might be the only possible A subgenome donor for *B. napus* formation, as the first domesticated *B. rapa*. Sampling of more wild *B. rapa* and *B. oleracea* relatives would be helpful for better understanding the complex events that occurred during *B. napus* origin. Noting prior evidence that the maternal ancestor of *B. napus* was highly unlikely to be *B. oleracea*[1], we posit that *B. napus* resulted from hybridization between a maternal ancestor of European turnip and a paternal common ancestor of the four aforementioned *B. oleracea* subspecies.

During the FSI, A subgenome-specific selection on defense-response genes greatly improved *B. napus* with regard to adaptation to environmental variation, reflecting the fact that strong survival pressure may often be the first challenge to a newly evolved plant species. Furthermore, the improvement of oil accumulation in *B. napus* resulted mainly from A subgenome-specific selection on triglyceride biosynthetic genes. This is similar to the history of upland cotton, whose D-subgenome-specific selection led to prolonged fiber elongation in cultivated cottons[47], suggesting that asymmetrical subgenomic selection may often occur during the FSI.

Artificial breeding of *B. napus* dramatically reduced its genetic diversity and affected SSI-selection signals. Combining this information with GWAS results and previous QTL information, we identified a number of outlier regions and candidate genes associated with seed yield, seed oil, glucosinolate, and erucic acid contents and flowering time. Similarly, improvement of seed quality due to subgenome parallel selection was also observed during the artificial selection of *B. napus*.

Our study provides a valuable resource for understanding the origin and improvement history of *B. napus* and will facilitate the dissection of the genetic bases of important agronomic complex traits. The significant SNPs associated with favorable variants, selection signals, and candidate genes will be a valuable resource for further improving the yield, seed quality, oil content, and adaptability of this recent allopolyploid crop and its relatives.

## Methods

**Plant materials**. The diversity panel used in this study comprised 588 *B. napus* accessions, including 466 from Asia, 102 from Europe, 13 from North America, and 7 from Australia (Supplementary Table 1; Supplementary Data 1). Based on growth habit records, these materials were divided into three ecotypes; spring (86 accessions), winter (74 accessions), and semi-winter (428 accessions). As there is no precise definition for the *B. napus* landrace, the *B. napus* accessions registered before 1980 and after 2000 were regarded as landraces and improved cultivars, respectively, for the purpose of investigating the selection signals during the FSI and SSI of *B. napus*. Detailed information of the 588 accessions is listed in Supplementary Table 1.

**Sequencing and discovery of genomic variations**. Prior to sequencing and phenotyping, all *B. napus* accessions were maintained by self-pollination for at least four generations at the Chongqing Rapeseed Engineering Research Center, Chongqing, China. Genomic DNA was extracted from the two youngest leaves on a single plant of each accession. DNA libraries with a mean insert size of 350 bp were constructed, and 125-bp paired-end reads were generated using an Illumina HiSeq 4000 instrument. Library preparation and sequencing were carried out at the Biomarker Technologies Corporation (Beijing, China). Two biological replicates for 20 accessions were selected for whole-genome resequencing to assess the accuracy of sequencing in this study.

The resequencing data for 199 *B. rapa* and 119 *B. oleracea* accessions published previously were retrieved from the National Center for Biotechnology Information (NCBI) database. Low-quality bases from paired-reads were trimmed using Trimmomatic (version 0.33)[48], with the parameter LEADING:3 TRAILING:3 SLIDINGWINDOW:4:15 MINLEN:80. To investigate the origin of *B. napus*, the genome sequences of *B. rapa* (version 1.5)[15] and *B. oleracea* (version 1.0)[16] were merged as a *B. napus* ancestral pseudo-genome. Filtered *B. napus* sequencing reads were aligned to the *B. napus* reference genome (version 4.1)[5] and the *B. napus*

ancestral pseudo-genome, and *B. rapa* and *B. oleracea* sequencing reads were mapped only to their corresponding reference genomes using the Burrows-Wheeler Aligner (version 0.7.10-r789)[49]. Local realignment, duplicate read marking, and base quality recalibration for the alignment results were further processed sequentially using Picard (release 2.0.1, http://broadinstitute.github.io/picard/) and GATK (version 3.2–2-gec30cee)[50]. SNP and InDel calling for each sample were performed using SAMtools's mpileup (version 0.1.19–44428cd) and GATK with the parameter–ts_filter_level 90. To remove false variants, only biallelic SNPs simultaneously detected by both methods were retained, using the function SelectVariants in GATK with the following parameters:–filterExpression QD < 2.0 || FS > 60.0 || MQ < 40.0–clusterWindowSize 5–clusterSize 2; SNPs with minor allele frequency (MAF) of lower than 5% in the population were discarded. A total of 103 SNPs were randomly selected for accuracy validation of SNP calling using conventional PCR and Sanger-based sequencing (Supplemental Table 5) in 20 *B. napus* accessions.

To reveal the original subtaxa of *B. rapa* and *B. oleracea* involved in the formation of *B. napus*, A (BraA) and C subgenome (BolC) SNP sets were obtained as the following procedure. We mapped the *B. napus* resequencing data onto the *B. napus* ancestral pseudo-genome to call SNPs, which were split into two SNP sets based on the two subgenomes. Then, the SNPs retrieved from the *B. rapa* or *B. oleracea* resequencing data using corresponding genomes as reference were combined with two SNP sets obtained previously to represent BraA and BolC SNP sets, respectively. To perform selective sweep analysis and GWAS in *B. napus*, the SNPs called from *B. napus* resequencing data (Bna) using the *B. napus* genome as reference were also obtained.

**Phylogenetic analysis and population structure analyses.** To construct maximum likelihood (ML) trees, we screened 17,000, 19,377, and 19,548 SNPs at fourfold-degenerate sites (MAF > 5%) from the BraA, BolC, and Bna SNP sets, respectively. We generated three ML trees using IQ-TREE v1.6.6[19], according to the best model determined by the Bayesian information criterion (BIC). The reliability of the ML trees was estimated using the ultrafast bootstrap approach (UFboot) with 1000 replicates. An online tool Interactive tree of life (iTOL) v3 (https://itol.embl.de)[51] was used to display the consensus ML trees. Principal component analysis of the three SNP sets was performed using EIGENSOFT (version 6.1.4)[52].

We inferred the population structure of 50 *B. napus* landraces and two progenitors, 199 *B. rapa* and 119 *B. oleracea* accessions, using STRUCTURE[53], with two subsets of SNPs (7668 and 10,396 SNPs with MAF>5% obtained from BraA and BolC, respectively; 1 SNP per 40 kb) evenly distributed throughout the reference genomes. Each K value, as a putative number of populations from 1 to 10, was obtained with five independent runs. The length of the burn-in period and number of MCMC replications after burn-in were set to 50,000 and 100,000, respectively. The optimum number of subgroups (K) was determined based on the log probability of the data (lnP(K)) and an *ad hoc* statistic ΔK method[54].

**Demographic history of *B. napus*.** To assess whether the *B. napus* C subgenome originated from the common ancestor of the four aforementioned *B. oleracea* subspecies, the *B. napus* C subgenome originated from the ancestor of European turnip, and whether and when migration into *B. napus* occurred, we investigated the demographic history using both diffusion approximations for demographic inference (∂a∂i) version 1.7.0[20] and fastsimcoal2 v. 2.6.0.3[21]. To mitigate the effect of LD, one SNP per 10 kb was selected, and SNPs located 10 kb away from genes were used to convert into a site frequency spectrum (SFS) using easySFS (https://github.com/isaacovercast/easySFS).

Five and seven models (Supplementary Figs. 3, 5) representing demographic history of *B. napus* A and C subgenomes were respectively evaluated using fastsimcoal2[21]. To ensure convergence, we ran fastsimcoal2 50 times, and kept the run that resulted in the highest log likelihood for each model. Estimates were obtained from 100,000 coalescent simulations per likelihood estimation (-n100,000) and 40 Expectation/Conditional Maximization cycle (-L40). The best model with the maximum likelihood was retained. We estimated 95% confidence intervals (CI) using parametric bootstrap estimates based on 80 data sets simulated according to Conditional Maximum Likelihood estimates in the best model estimation parameters. The optimal demographic models obtained from fastsimcoal2 were also validated using ∂a∂i[20]. Since up to three subpopulations can be handled in ∂a∂i at a time, three demographic models for both *B. napus* two subgenomes were compared. The best fitting model was selected with the highest log-likelihood under the best-fit parameter set.

To better understand the original form of *B. napus*, we used SMC++[22] to estimate the divergence time and historical *Ne* among different ecotype and usage of *B. napus*. In the demographic analyses, generation estimates were inferred by assuming that the upper and lower mutation rates were $1.5 \times 10^{-8}$ and $9 \times 10^{-9}$ per synonymous site per generation, respectively, and that the generation time was 1 year[46].

**Calculation of linkage disequilibrium.** Linkage disequilibrium (LD) between pairs of SNPs up to 100 kb apart on each chromosome were estimated as the correlation coefficient ($r^2$) using PLINK v1.07[55], LD decay statistics were calculated for

different (sub)populations, and LD decay graphs were plotted using PopLDde-cay3.26 (https://github.com/BGI-shenzhen/PopLDdecay), with the parameter -MaxDist 1000 for SNP set BraA, and -MaxDist 5000 for SNP sets BraC and Bna.

**Measurement of agronomic traits.** All accessions evaluated in this study were cultivated in field trials in Beibei, Chongqing, China (29°45′ N, 106°22′ E, 238.57 m) during the 2013 and 2014 growing seasons. A randomized complete block design with three replications was employed. Each accession was grown in three rows of 10 plants per row, with a distance of 20 cm between plants and 30 cm between rows. Open-pollinated seeds were harvested from five randomly selected plants for measurements of seed yield per plant (SY) and silique length (SL). Oil content (OC), seed color (SC), and total glucosinolate content (GC) were detected by near-infrared reflectance spectroscopy (NIRS) analysis. The fatty acids palmitic acid, stearic acid, oleic acid, linoleic acid, linolenic acid, eicosenoic acid, and erucic acid were quantified by gas liquid chromatography, according to a previous report[26]. The flowering time (FT) was defined as the day on which the plants reached Biologische Bundesanstalt, Bundessortenamt and CHemical (BBCH) stage 61[56].

**Genome-wide selective sweep analysis.** To identify candidate regions potentially affected by selections, nucleotide diversity (π), and population fixation statistics ($F_{ST}$) were calculated using vcftools (v0.1.13, https://vcftools.github.io) in a 100-kb sliding window with a step size of 10 kb. π is the expected heterozygosity per site derived from the average number of sequence differences in a group of samples, and $F_{ST}$ is the inbreeding coefficient of a subpopulation, used for estimating the degree of pairwise genomic differentiation on candidate genes between pairs of subpopulations. The reduction of diversity (ROD) values were calculated based on the ratio of π for a subpopulation with respect to a control subpopulation. All the output results of ROD and $F_{ST}$ were standardized and transformed into z-scores using a 100-kb sliding window with a 10-kb step size. The outlier windows of ROD (high values), and $F_{ST}$ (high values), which characterize candidate genes involved in the FSI and SSI of *B. napus*, were determined by z-tests with a significance level of α = 0.05 corresponding to a z-score of 1.645.

To identify additional selection effects, we also calculated the XP-CLR and cross-population extended haplotype homozygosity (XP-EHH) using XP-CLR v1.0 (–w1 0.005 200 10000–p0 0.95)[57] and selscan v1.2.1[58], respectively. In the two programs, all the Bna SNPs were assigned genetic positions based on a published genetic map. Windows with the top 5% of maximum XP-CLR or XP-EHH values were considered as selection regions. Candidate genes involved in the domestication *B. napus* were identified from the selection regions detected by at least two methods, and those for SSI were identified from common selection regions detected by at least three methods.

In the FSI-selection analysis, European turnip (n = 33) and *B. oleracea* (n = 66) involved in the original formation of *B. napus* versus *B. napus* landraces and the SNP sets BraA and BolC were used to detect FSI events. The improvement signals were detected based on the Bna SNP set and the following three pairwise comparisons: first, *B. napus* landraces versus improved cultivars to identify SSI events; second, double-low versus double-high *B. napus* accessions to identify selection signals related to seed quality improvement; third, comparisons between spring and winter ecotype, and between spring and winter ecotype to identify selection signals associated with ecotype improvement.

**Genotype imputation and accuracy estimation.** In the case of low-depth sequencing (~5×), genotyping calling errors are inevitable, particularly at sites of low MAF. To improve the mapping resolution for GWAS, imputation for sporadic missing genotypes in the Bna SNP set was implemented in Beagle v.4.1[59] with default parameter settings and 50 iterations in a sliding window of 50,000 SNPs. Pedigree and reference information were not used in the imputation procedure. Imputation accuracy was estimated using two measures. One is the comparison of the imputation results of 19 polymorphic nucleotides with the corresponding Sanger sequencing results (Supplementary Data 32). The other was the correlations ($r^2$) between imputed and true genotypes, which were calculated at each locus for 20 accessions of biological replicates (R021–R040, Supplementary Data 1) in intervals of 5% of MAF. Missing SNPs in the true genotypes were excluded when calculating the correlations.

**Genome-wide association study.** The best linear unbiased prediction (BLUP) for each investigated trait of each accession was obtained using an R script (www.eXtension.org/pages/61006), based on a linear model and corresponding traits with three replicates over 2 years. The resulting values were used as phenotypes for the association analysis. The multi-locus random-SNP-effect mixed linear model was used to test trait-SNP associations in mrMLM v1.3[24]; PCA and K were controlled as fixed and random effects, respectively. The effective number (n) of independent SNPs was calculated using GEC software[25]. Significant (0.05/n, Bonferroni correction) and suggestive (1/n) and P-value thresholds were set to control the genome-wide type I error rate. To identify reliable significant association signals in our GWAS, only LD blocks containing at least one significant and one suggestive SNPs were regarded as significant loci.

**Transcriptome and gene ontology enrichment analyses**. Total RNA was isolated from 11 tissues of two representative accessions with different oil contents and seed qualities (Zhongshuang11, a high-oil-content, double-low accession; and Zhongyou821, a low-oil-content, double-high accession). Tissues from roots (Ro), stems (St), mature leaves (Le), and flowers (Fl) were harvested at flowering stage 63–65 as defined in the BBCH[56]. Seeds (Se) and silique pericarps (SP) on the main inflorescence were harvested 7, 10, 15, and 45 days after flowering. For each sample, two biological replicates, each replicate obtained from three independent plants, were pooled for transcriptome sequencing.

Library preparation, sequencing, and read filtering of 44 libraries (11 tissues × two accessions × two biological replicates per sample) were conducted by the Biomarker Technologies Corporation (Beijing, China). Filtered reads were then mapped to the *B. napus* reference genome using STAR 2.4.2a with default parameters[60]. Gene expression levels were quantified by the cuffquant program in terms of fragments per kilobase of transcript per million mapped reads (FPKM)[61]. The DEGs were then detected in each tissue between two accessions using the cufdiff program, with the following standards: FDR < 0.01 and absolute fold change >2.

**Gene ontology enrichment analysis**. Gene ontology (GO) terms for all *B. rapa*, *B. oleracea*, and *B. napus* proteins were assigned based on BLASTP analysis against the entire *A. thaliana* proteome, with an e-value cut-off of 1e-5 and max_target_seqs of 1. The R package clusterProfiler was then used to identify significantly enriched GO terms (false discovery rate (FDR) < 0.05)[62]. The resulting *P*-values were corrected for multiple comparisons using the method of Benjamini and Hochberg. To avoid overestimation, tandem-duplicated genes in the outlier regions were filtered before the GO enrichment analysis.

**Reporting Summary**. Further information on experimental design is available in the Nature Research Reporting Summary linked to this article.

## Data availability

The raw sequencing data have been deposited in the NCBI database under BioProject accession codes PRJNA358784 and PRJNA430009. The raw sequencing data have also been deposited in the BIG Data Center under BioProject accession codes PRJCA000376 and PRJCA001246. The source data underlying Figs. 1–5 are provided as a Source Data file. Data supporting the findings of this work are available within the paper and its Supplementary Information files. A reporting summary for this Article is available as a Supplementary Information files. The datasets generated and analyzed during the current study and the plant materials used in this study are available from the corresponding author upon reasonable request.

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

## Acknowledgements

We thank Professor William Lucas for critical reading of the manuscript. This work was supported by grants from the National Natural Science Foundation of China (31830067, U1302266, 31571701, and 31871653), the 973 Project (2015CB150201), the National Key Research and Development Plan (2018YFD0100501 and 2016YFD0101007), the 111 Project (B12006), the Natural Science Foundation of Chongqing (cstc2018jcyjA1219).

## Author contributions

J.L., X.W., and A.H.P. designed and supervised the study; K.L., X.L., L.W. and J.W. collected the samples and performed the phenotyping; M.L., C.Z., Z.C., Z.X., H.J., K.Z., H.D. and C.Q. measured the agronomic traits; K.L. and L.W. led the data analysis with contributions from W.Q., L.L., R.W., Q.Z., J.Y., Y.F., W.S. and X.X. and Y.W., F.C., X.C., J.M., Y.L., Y.-R. C., Y.H., Z.T., H.W., Y.N., N.-N.L., G.Z., H.Z. and N.L. contributed to the data collection and interpretation of results, and K.L., X.W., A.H.P. and J.L. drafted the paper. All authors reviewed the manuscript.

## Additional information

**Competing interests:** The authors declare no competing interests.

