## [Peer Review File · Nature Communications]

Reviewers' comments:

Reviewer #1 (Remarks to the Author):

The authors sequenced 588 *B. napus* accessions, 199 *rapa* and 119 *oleracea* accessions. They called 5,294,158 SNPs, some indels and CNVs. The SNPs were used to query the history of canola. They couldn't find the progenitor of *B. napus*' C-genome, but did find some likely candidates for the A-genome in Oilseed/Asian turnips. Overall it is an interesting manuscript, however there is room for some improvement.

They looked at regions of selection using reduction of diversity (ROD) and fixation index (Fst). This is valid, but more detailed population genetics analysis could be done with this data.

The genes in the ROD/Fst outlier regions could have been characterised in more detail, the discussion only looks at GO-terms enriched in these regions.

The SNPs, CNVs and indels were used for GWAS using 12 phenotypes. The stepwise regression to find the explanatory power of SNPs/CNVs does not seem to account for population structure. If alternative methods were used, these very high percentages of explained phenotype variance (78%, 69%) will probably reduce.

RNASeq was used to look for differentially expressed genes between a high and a low oil content canola. Could the authors explain why only two biological replicates were used instead of the more commonly used three?

It would have been good to compare transposon activity in the *napus* genome compared with the *rapa/oleracea* genome, but the coverage in all individuals may be too low

Reviewer #2 (Remarks to the Author):

The work of Lu et al. is the most comprehensive work yet on the population genetics of *B. napus*.

In terms of novelty, I identified another *B. napus* pop gen paper that was not cited and likely should be to provide a clear picture of the landscape.

Gazave et al. 2016. Population Genomic Analysis Reveals Differential Evolutionary Histories and Patterns of Diversity across Subgenomes and Subpopulations of *Brassica napus* L. *Frontiers in Plant Science* <https://www.frontiersin.org/article/10.3389/fpls.2016.00525>

Given the absence of true wild accessions, it seems that the study can only be most confidently focused on crop improvement, thus I would suggest that domestication should be deleted from the title and keep improvement, or call it post-domestication.

Lines 162-166, there are multiple evolutionary forces acting to shape LD patterns in the genome and it is certainly not conclusive to assume that a more rapid rate of LD decay reflects hybridization signals. More lines of evidence are needed than LD alone.

Line 234, the selection of a MAF of 0.03 for GWAS should be justified and other analyses performed to demonstrate that the lower frequency variants adhere to a uniform p-value distribution and not have spuriously inflated P-values. It is best to have at least 25-30 individuals with the minor allele to accurately calculate a trait mean for that group.

Lines 238-247, the LD decay relative to the peak SNPs needs to be articulated in this section and in the Figure. The mapping resolution or rate of LD decay where candidate genes are identified needs to be provided. The concordance with previous studies need to also be anchored based on LD.

Lines 610-616, the overall imputation accuracy and for each subpopulation should be reported.

Line 624-627, the Type I error rate is better set at 0.05 B. H. FDR, or with the simpleM approach that takes into account of linkage disequilibrium of tested variants.

Lines 629-635, the model fitting needs to be done with PCs and Kinship. It is better to use a genome-wide multilocus mixed model (MLMM) that uses stepwise regression with PCs and kinship on both separate classes of ALL variants at once instead of fitting a stepwise regression model that is agnostic to the genome and does not control for population structure and relatedness.

Line 648, what was pooled – 3 independent plant or all 6 plants (3 independent plants x 2 biological replicates). That needs to be more clearly written.

Supplementary Figure 10, the Type I error rates for some traits in GWAS are not well controlled for as shown in the Q-Q plots. It seems some of this is massive long-range LD based on the Manhattan plots (i.e., low mapping resolution) and other issues of still residual uncontrolled population structure for flowering time.

Reviewer #3 (Remarks to the Author):

The manuscript describes an analysis of *Brassica napus* through whole-genome re-sequencing. The main claims are that the likely progenitors of the allopolyploid *B. napus* are probably European turnip (contributing A genome) and a common ancestor of four *Brassica oleracea* morphotypes (contributing C genome). In addition by identifying selective sweeps for SNPs in "landraces" and "improved" *B. napus* lines the authors suggest candidates genes that have contributed to the adaptation of *B. napus*.

The manuscript is generally well written and the analyses appear to have been extremely thorough. My main concern is the complete omission of the fact that semi-winter *B. napus* (which represented ~80% of the lines studied) was generated through the introgression of *B. rapa* (A genome) alleles into the winter *B. napus* background (eg. Qian et al, 2006, there are multiple references for this). A number of the results presented which relate to more variation being observed in the A genome is undoubtedly relate to the known derivation of semi-winter types, yet this is never referred to. One would also suspect that the introgression of *B. rapa* genomic regions into *B. napus* might impact efforts to identify the progenitors of *B. napus*. Yet, the authors do appear to have confirmed previous reports that European turnip is the most likely A genome progenitor. The fact that the bulk of the analyzed lines are semi-winter types (~80%) does somewhat skew the analysis of the *B. napus* genotypes, but it is noted that the authors have tried to assuage this problem by limiting some of the analyses to only 50 "landrace" lines. The introgression of A genome segments has probably also led to some of the suggested asymmetrical sub-genome evolution, in particular in the context of the ecotype improvement discussed on page 17.

There is also no mention of the fact there is preferential replacement of C genome regions with the A genome in *B. napus* through homoeologous recombination events (HE) (Chalhub et al, 2014), which is probably reflected in some of the differences that are observed between the A and C genomes. Interestingly the authors suggest that they looked for HE in their data (based on M&M) but do not refer to this analysis in the main text. Although their depth of sequencing (average ~5x) is probably too low to identify such events, so perhaps this should be removed from the methods, but the known prevalence of such events should be mentioned in the context of genome adaptation. For example,

such events have been shown to impact flowering time and seed glucosinolate content.

With respect to the analyses completed, some elements do need to be clarified. It is not clear that the SNP numbers that are presented (eg. 733,165 BraA and 1,095,281 BolC) represent SNPs that would be informative across the whole dataset. It is certain that there would have been SNPs that were only informative among the *B. rapa* and the *B. napus* genotypes, respectively (similarly for the C genome). But if the authors only used a cut-off criteria of an allele frequency $>1\%$ or >0.03 (both values are indicated in M&M) across the whole population, which presumably means both *B. rapa* and *B. napus* in the case of the A genome, then SNPs that were specific to *B. rapa* and/or *B. napus* would be retained.

The lengthy discussion of potentially important candidate genes that have contributed to the adaptation of *Brassica napus* has not really identified many novel candidates, most have been suggested in previous publications; however, the authors did present corroborating GWAS data, which certainly strengthened this section compared to some previous work.

Minor points:

P9, there were probably insufficient fodder and vegetable types represented in the dataset to make any comments on differential LD in these lines.

P10, suggesting the C genome has lost diversity in comparison to the A genome, contradicts all evidence that the A genome diversity has been increased by multiple introgressions.

P17, the references used, no. 25 and 26, do not appear to support the statements.

P6 and P25, there is a contradiction between the main text and the M&M, the main text indicates 103 SNPs were validated and the M&M only 20.

P28, "To mitigate the effect of LD, one SNP per 10Kb was selected.." – Based on all previous publications and the authors own data this distance would be grossly insufficient to mitigate the effect of LD.

Reviewer #4 (Remarks to the Author):

In this manuscript, the authors describe the resequencing and analysis of 588 diverse *Brassica napus* (rapeseed) accessions aimed at investigating the parental origins, at the morphotype level, of this important allopolyploid crop. The origin of this crop remains poorly understood, in large part due to its polyploid nature and complex history of admixture during domestication. Furthermore, the progenitor species (*B. rapa* and *B. oleracea*) each have their own complex evolutionary histories riddled with polyploidy, hybridization and introgression events. The authors also identified genes associated with important agricultural traits using a combination of genomic and transcriptomic analyses. These findings and new resources have the potential to be a valuable resource to the community for future genomic studies and to develop tools to guide future breeding efforts.

However, after reviewing the methods, I have several major concerns:

1. A Neighbor-Joining (NJ) method was used to estimate relationships among resequenced accessions combined with publicly available data for progenitor species in an attempt to identify which morphotypes contributed to the origin of *B. napus*. This is a distance based approach, not a phylogenetic method, and typically results in estimates that are highly incongruent with a more rigorous likelihood methods for inferring phylogenies. Furthermore, the methods used by the authors lack any statistical power. I strongly recommend that the authors reanalyze the data with a more rigorous phylogenetic method that uses an evolutionary model; maximum-likelihood approach (GTR+G Model) with bootstrapping to estimate node support. In short, the results from the current "phylogenetic" analyses are questionable and impact downstream analyses.

2. This study lacks sampling of any wild A subgenome subspecies e.g. field mustard (*B. rapa* subsp. *sylvestris*). Thus, the authors should not conclude that European turnip types are the A subgenome ancestor contributed to present day *B. napus*. I suggest that the authors either modify the text to reflect this taxon sampling problem or reanalyze the data with additional sampling of wild A subgenome species (See Guo et al. 2014 or Qi et al. 2016). It is possible that the appropriate lineages just weren't sampled. Also see below minor comment 4)

Guo, Y., Chen, S., Li, Z., & Cowling, W. A. (2014). Center of origin and centers of diversity in an ancient crop, *Brassica rapa* (turnip rape). *Journal of Heredity*, 105(4), 555-565.

Qi, X., An, H., Ragsdale, A. P., Hall, T. E., Gutenkunst, R. N., Chris Pires, J., & Barker, M. S. (2017). Genomic inferences of domestication events are corroborated by written records in *Brassica rapa*. *Molecular ecology*, 26(13), 3373-3388.

3. Lines 146-148. "..., our data support the model indicating that the ancestor of *B. napus* split from the common ancestor of four *B. oleracea* subspecies at about 7,500–12,500 years ago, ...". These date estimates predate the origin of all the *B. oleracea* morphotypes. The origin of *B. napus*, if it only involved these four subspecies (i.e. morphotypes), obviously could not have occurred prior to the domestication of *B. oleracea* (i.e. origin of all cruciferous vegetables). This is likely due to the methods used to estimate relationships (see #1 above). Also, strong bootstrap support values are needed to confidently identify progenitor species.

4. Lines 199-201. "... the C subgenome of *B. napus* might be caused by the multiplicity of origins of that subgenome." See below minor comment #1. I suspect that multiple origins of *B. napus* is far more likely than a single origin; given that hybridization between these progenitor species and polyploidization can spontaneously happens in gardens and greenhouses.

5. I would be hesitant to estimate copy number variation with only 3-7X sequence depth especially in a highly polyploid species. There is just too much variation in coverage at this sequence depth to accurately call presence-absence or even copy number variation. I would recommend removing this from your analyses. Have you experimentally validated any of these 3,639 CNVs (lines 88-89)?

Additional minor comments and suggestions:

1. Lines 62-65 "The precise identities of the two progenitors that hybridized to form *B. napus* remain elusive, as *B. rapa* and *B. oleracea* occur in geographically distinct areas and each has morphologically diverse subspecies." This is not entirely correct. These species have been commonly cultivated together throughout Europe for hundreds of years. Hybridization between these species is common, and hybrids are known to spontaneously double to polyploids. I've personally observed this in the greenhouse. This is why present day *B. napus* diversity is unlikely a single origin.

2. Why was flowering time further investigated given that it wasn't highly enriched in the GO term enrichment analysis?

3. Lines 73-75: "In this study, we performed deep genome sequencing of 588 *B. napus* accessions and transcriptome sequencing of 11 tissues from two *B. napus* accessions with different seed quality." 3-8x sequence depth is not "deep genome sequencing". This would be considered "genome skimming" or "shallow genome sequencing".

4. To support claims of a turnip (*B. rapa* ssp. *rapa*) origin for the A subgenome, further evidence should be provided that domesticated turnips existed during time of *B. napus* formation (7,500-12,000 years ago). Highlighting archaeological or linguistic evidence would greatly strengthen this argument

(e.g. Ignatov et al. 2008 and Reiner 1995).

Ignatov, A. N., Artemyeva, A. M., & Hida, K. (2008, September). Origin and expansion of cultivated *Brassica rapa* in Eurasia: linguistic facts. In V International Symposium on Brassicas and XVI International Crucifer Genetics Workshop, Brassica 2008 867 (pp. 81-88).

Reiner, H., Holzner, W., & Ebermann, R. (1995). The development of turnip-type and oilseed-type *Brassica rapa* crops from the wild-type in Europe—An overview of botanical, historical and linguistic facts. *Rapeseed Today Tomorrow*, 4, 1066-1069.

Responses to reviewers' comments

Reviewers' comments:

Reviewer #1 (Remarks to the Author):

The authors sequenced 588 *B. napus* accessions, 199 *B. rapa* and 119 *B. oleracea* accessions. They called 5,294,158 SNPs, some indels and CNVs. The SNPs were used to query the history of canola. They couldn't find the progenitor of *B. napus*' C-genome, but did find some likely candidates for the A-genome in Oilseed/Asian turnips. Overall it is an interesting manuscript, however there is room for some improvement.

1. They looked at regions of selection using reduction of diversity (ROD) and fixation index (F_{ST}). This is valid, but more detailed population genetics analysis could be done with this data.

RESPONSE:

Thanks for your valuable suggestion. In the original manuscript, only ROD and F_{ST} were used to identify domestication- and improvement-selection signals. To increase the reliability of the domestication-selection results, we combined the selection signals of ROD and F_{ST} , and only overlapping outlier regions were regarded as true selection signals. In addition to ROD and F_{ST} , we detected improvement-selection signals using two other methods, XP-CLR (cross-population composite likelihood ratio test) and XP-EHH (cross-population extended haplotype homozygosity). The selection outlier regions were obtained when they were simultaneously detected by at least three of the four methods (F_{ST} , ROD, XP-CLR, and XP-EHH). As shown in Supplementary Figs. 14, 15, 29, and 30 in the revised manuscript, several improvement-selection signals were detected repeatedly using

different methods. These results increase the reliability of newly identified selection signals, which is very helpful for our further investigations.

2. The genes in the ROD/Fst outlier regions could have been characterised in more detail, the discussion only looks at GO-terms enriched in these regions.

RESPONSE:

Thank you for the good suggestion. We have made discussion about the genes in the outlier regions, especially genes improving environmental adaptation during *B. napus* domestication and those involved in ecotype improvement of *B. napus*. For example, we found that “genes enriched in GO terms associated with defense responses may play key roles in local environmental adaptation of *B. napus*. In the A subgenome, we identified 6, 5, 6 and 14 genes associated with defense response to drought tolerance, herbivore resistance, responses to mechanical stimulus, and immune responses, respectively (Supplementary Table 31)”. Furthermore, we included a discussion on ABA biosynthetic genes and disease-responsive genes, and their potential contribution to local environmental adaptation during *B. napus* domestication:

“During domestication, genes enriched in GO terms associated with defense responses may played key roles in local environmental adaptation of *B. napus*. In the A subgenome, we identified 6, 5, 6 and 14 genes associated with defense response to drought tolerance, herbivore resistance, responses to mechanical stimulus, and immune responses, respectively (Supplementary Table 31). Among the defense-responsive genes involved in drought tolerance, *NCED3* (*Bra001552*) might well be the most interesting one, as it has been shown to play a major role in the regulation of ABA (abscisic acid) biosynthesis in response to water deficit²⁷. Another drought tolerance gene, *NCED5* (*Bra032359*) contributes, along with *NCED3*, to ABA production, thereby affecting plant growth and water stress tolerance²⁸.

Endogenous ABA is rapidly produced during drought, inducing stomatal closure and thereby enhancing adaptation capacity against drought stress. A

subgenome-specific selection on ABA biosynthesis pathway genes might have been important for enhancing drought tolerance during *B. napus* domestication and laid a solid foundation for *B. napus* cultivation in diverse environments. Other candidate genes involved in ABA metabolism, in the selection regions, are also noteworthy, such as *CYP707A3* (*Bra021965* and *Bra025083*), *XERICO* (*Bra013211*) and *PHYB* (*Bra001650*), all of which are associated with drought stress responses via regulation of ABA accumulation (Supplementary Table 31).

In addition to drought tolerance, A subgenome-specific selection also contributed to disease resistance improvement during *B. napus* domestication. In the selection regions, *NPR3* (*Bra025093*) is one of the notable genes. As a receptor for the immune signal salicylic acid (SA) in plants, NPR3 controls the proteasome-mediated degradation of NPR1, which is involved in negative regulation of defense responses against bacterial and oomycete pathogens, in a SA-regulated manner²⁹. Similar defense-responsive genes, *BAH1* (*Bra032581*), *NHL25* (*Bra028103*), and *NDR1* (*Bra035766*) are also involved in regulating plant innate immunity to microbes (Supplementary Table 31) and may have contributed to biotic stress during *B. napus* domestication.”

3. The SNPs, CNVs and indels were used for GWAS using 12 phenotypes. The stepwise regression to find the explanatory power of SNPs/CNVs does not seem to account for population structure. If alternative methods were used, these very high percentages of explained phenotype variance (78%, 69%) will probably reduce.

RESPONSE:

Thank you for raising this important point. In the original manuscript, we estimated the explanatory power of significant SNPs using stepwise regression analysis without controlling for population structure, and this led to an overestimation of the total phenotype variance explained. To avoid this problem, we conducted multi-locus random mixed linear model analysis for GWAS using mrMLM, which could significantly increase the statistical power and decrease Type 1 errors compared with other methods. Though the number of SNPs significantly associated with our

target traits and the explained phenotype variance (14.30% to 35.47%) was decreased in the mrMLM analyses, the accuracy of the GWAS results was more reliable than that in the original manuscript.

In addition, we have deleted the GWAS results for CNVs, following the suggestion of other reviewers, because the depth of resequencing (~5.5×) was insufficient to identify the CNVs accurately.

4. RNASeq was used to look for differentially expressed genes between a high and a low oil content canola. Could the authors explain why only two biological replicates were used instead of the more commonly used three?

RESPONSE:

It is true that three biological replicates are commonly used in transcriptome analyses. In our study, two biological replicates were used instead of the more commonly used three, because the expression patterns have been analyzed in different tissues at different stages of development in another transcriptome analysis project. We harvested more than 110 samples, including 16 different organs at different growth stages. For example, seeds were collected at 3, 5, 7, 13, 19, 21, 24, 27, 30, 35, 40, 43, 46, and 49 days after flowering. Most correlation coefficients between two biological replicates were higher than 0.95, and time series samples and the same organs at different stages also showed high correlation coefficients. In addition, the results of transcriptome sequencing have been widely validated by qRT-PCR in our previous studies, suggesting that our transcriptome analysis results are reliable, and suitable for identifying differentially expressed genes.

5. It would have been good to compare transposon activity in the napus genome compared with the rapa/oleracea genome, but the coverage in all individuals may be too low

RESPONSE:

Thank you for the good suggestion. Due to the insufficient sequencing depth, it is hard to identify the transposons and compare their activity. We might compare transposon activity in the accessions with high coverage in further investigations.

Reviewer #2 (Remarks to the Author):

The work of Lu et al. is the most comprehensive work yet on the population genetics of *B. napus*.

In terms of novelty, I identified another *B. napus* pop gen paper that was not cited and likely should be to provide a clear picture of the landscape.

Gazave et al. 2016. Population Genomic Analysis Reveals Differential Evolutionary Histories and Patterns of Diversity across Subgenomes and Subpopulations of *Brassica napus* L. *Frontiers in Plant Science* <https://www.frontiersin.org/article/10.3389/fpls.2016.00525>

1. Given the absence of true wild accessions, it seems that the study can only be most confidently focused on crop improvement, thus I would suggest that domestication should be deleted from the title and keep improvement, or call it post-domestication.

RESPONSE:

Thank you for your suggestion. Domestication is the process of artificial selection that leads to wild plants becoming cultivated landraces. So far, no truly wild populations of *B. napus* have been identified, and it is impossible to investigate the domestication process without a wild population using traditional methods. To solve the problem, we firstly identified the two progenitors that hybridized to form *B. napus* and found that the *B. napus* A and C subgenomes might be derived from the ancestor of European turnip and the common ancestor of four *B. oleracea* subspecies, respectively. Then, we pooled two groups of progenitors, European turnip ($n = 33$) and four *B. oleracea* subspecies ($n = 66$), to represent the pseudo-wild ancestral A and C subgenomes of *B. napus*, respectively. Hence, we could compare the two

pseudo-wild ancestral subgenomes of *B. napus* and the corresponding subgenomes in *B. napus* landraces, to identify domestication selection signals. Although the method is not perfect, it is still a good attempt to investigate the domestication events for those species that have no known wild ancestors. Thus, we would like to keep the domestication analysis results in the revised manuscript.

2. Lines 162-166, there are multiple evolutionary forces acting to shape LD patterns in the genome and it is certainly not conclusive to assume that a more rapid rate of LD decay reflects hybridization signals. More lines of evidence are needed than LD alone.

RESPONSE:

Thanks for your valuable suggestion. We agree that we cannot conclusively state that a more rapid rate of LD decay reflects hybridization signals. We also used SMC++ to estimate the historical effective population sizes (N_e) and divergence times for different *B. napus* ecotypes. The following paragraph has been added to the revised manuscript.

“The winter and semi-winter *B. napus* ecotypes diverged ~60 years ago, whereas the winter and spring *B. napus* diverged ~400 years ago, and oilseed and non-oilseed *B. napus* diverged ~276 years ago. These results are consistent with historical records, which indicate that spring *B. napus* developed ~1700 years ago and spread to England in the late 18th century¹⁹, and that the semi-winter ecotype has only a short history in China, and arose from the winter ecotype, which was introduced from Europe in the 1930–1940s and adapted to the local environment²⁰. Based on the LD and demographic analyses, and on a literature survey, we speculate that winter oilseed was the original *B. napus*.”

3. Line 234, the selection of a MAF of 0.03 for GWAS should be justified and other analyses performed to demonstrate that the lower frequency variants adhere to a uniform p-value distribution and not have spuriously inflated P-values. It is best to

have at least 25-30 individuals with the minor allele to accurately calculate a trait mean for that group.

RESPONSE:

Thanks for your suggestion. We have filtered all the SNPs with a MAF of > 0.05 and obtained 529,771 675,457, and 670,028 high quality SNP sets in the three SNP sets BraA, BolC, and Bna. All the SNPs were used for phylogenetic analyses, domestication and improvement selection signal detection, and demographic inferences of *B. napus* evolutionary history, and SNPs in Bna were only used for GWAS in *B. napus*. In this study, a total of 588 *B. napus* accessions were used, which ensured that there were at least 30 individuals in a group when MAF was set to 0.05. Based on the new SNP data, we have repeated the phylogenetic, selection, and demographic analyses. All the results have been updated in the revised manuscript.

The results of the population structure analyses were retained, since the SNPs were selected with a MAF of 0.05.

4. Lines 238-247, the LD decay relative to the peak SNPs needs to be articulated in this section and in the Figure. The mapping resolution or rate of LD decay where candidate genes are identified needs to be provide. The concordance with previous studies need to also be anchored based on LD.

RESPONSE:

Thanks for your valuable suggestion. We have calculated the LD decay at the whole genome level and improved our results. The LD blocks containing peak SNPs were provided in the revised Figures 4 and 5. The concordance with previous studies were compared based on the LD blocks.

5. Lines 610-616, the overall imputation accuracy and for each subpopulation should be reported.

RESPONSE:

We have added a description of the imputation accuracy in the revised manuscript. The following two sentences “Imputation accuracy was estimated by

comparing the imputation results of 19 polymorphic nucleotides with the corresponding Sanger sequencing results (Supplementary Table 34).” and “To obtain high-quality SNPs, we performed imputation for the Bna SNP set, and retained 670,028 SNPs with a MAF of > 5% for GWAS. Comparison between imputation results for 19 polymorphic nucleotides and Sanger sequencing results indicated that 98.74% of imputed genotypes were correct (Supplementary Table 27).” have been added to the Methods and Results sections, respectively.

6. Line 624-627, the Type I error rate is better set at 0.05 B. H. FDR, or with the simpleM approach that takes into account of linkage disequilibrium of tested variants.

RESPONSE:

Thanks for your suggestion. The Benjamini and Hochberg FDR method could control the genome-wide type I error rate better than Bonferroni correction, and is very powerful in GWAS of crop quality traits. However, this method is too stringent for complex traits, and Bonferroni correction has also often been used in GWAS, especially in studies of complex traits. In this study, we changed the traditional MLM model to the multi-locus random-SNP-effect MLM model using mrMLM, which could significantly increase the statistical power and decrease Type 1 error. Hence, in our results, significant ($0.05/n$, Bonferroni correction) and suggestive ($1/n$) values and *P*-value thresholds were set to control the genome-wide type I error rate. To identify reliable significant association signals in our GWAS, only LD blocks containing at least one significant and one suggestive SNP were regarded as significant loci. The results showed that the majority of significant loci were in accordance with the QTLs detected in previous studies, and several important genes with established functions were located within the LD blocks, such as flowering time gene *FLC* on chromosome C02, suggesting that the type I error rate has been controlled in our GWAS results.

7. Lines 629-635, the model fitting needs to be done with PCs and Kinship. It is better to use a genome-wide multilocus mixed model (MLMM) that uses stepwise regression with PCs and kinship on both separate classes of ALL variants at once

instead of fitting a stepwise regression model that is agnostic to the genome and does not control for population structure and relatedness.

RESPONSE:

We appreciate this comment. In the revised manuscript, we have repeated the GWAS using the multi-locus random-SNP-effect mixed linear model (MLM) program mrMLM v1.3, which could significantly increase the statistical power and decrease Type 1 errors compared with other methods. Considering the PCs and kinship, the total phenotype variance explained by significant SNPs decreased and ranged from 14.30% to 35.47% in the mrMLM analyses (Supplementary Table 28).

8. Line 648, what was pooled – 3 independent plant or all 6 plants (3 independent plants x 2 biological replicates). That needs to be more clearly written.

RESPONSE:

Thank you for pointing out this problem. We have changed the description as follows: “For each sample, two biological replicates, each replicate obtained from three independent plants, were pooled for transcriptome sequencing.” in the revised manuscript.

9. Supplementary Figure 10, the Type I error rates for some traits in GWAS are not well controlled for as shown in the Q-Q plots. It seems some of this is massive long-range LD based on the Manhattan plots (i.e., low mapping resolution) and other issues of still residual uncontrolled population structure for flowering time.

RESPONSE:

Thank you for raising this concern. Due to the Type I error rates for some traits in our GWAS results, we changed the GWAS model from the traditional MLM model to the multi-locus random-SNP-effect mixed linear model using mrMLM v1.3. The Manhattan and QQ plots in the revised manuscript showed that the Type 1 error has been well controlled, especially for flowering time (Supplementary Fig. 16).

Reviewer #3 (Remarks to the Author):

The manuscript describes an analysis of *Brassica napus* through whole-genome re-sequencing. The main claims are that the likely progenitors of the allopolyploid *B. napus* are probably European turnip (contributing A genome) and a common ancestor of four *B. oleracea* morphotypes (contributing C genome). In addition by identifying selective sweeps for SNPs in “landraces” and “improved” *B. napus* lines the authors suggest candidates genes that have contributed to the adaptation of *B. napus*.

1. The manuscript is generally well written and the analyses appear to have been extremely thorough. My main concern is the complete omission of the fact that semi-winter *B. napus* (which represented ~80% of the lines studied) was generated through the introgression of *B. rapa* (A genome) alleles into the winter *B. napus* background (eg. Qian et al, 2006, there are multiple references for this). A number of the results presented which relate to more variation being observed in the A genome is undoubtedly relate to the known derivation of semi-winter types, yet this is never referred to. One would also suspect that the introgression of *B. rapa* genomic regions into *B. napus* might impact efforts to identify the progenitors of *B. napus*. Yet, the authors do appear to have confirmed previous reports that European turnip is the most likely A genome progenitor. The fact that the bulk of the analyzed lines are semi-winter types (~80%) does somewhat skew the analysis of the *B. napus* genotypes, but it is noted that the authors have tried to assuage this problem by limiting some of the analyses to only 50 “landrace” lines. The introgression of A genome segments has probably also led to some of the suggested asymmetrical sub-genome evolution, in particular in the context of the ecotype improvement discussed on page 17.

RESPONSE:

Thanks for the comments. We have realized the potential impacts on the progenitor identification due to the introgression of *B. rapa* and *B. oleracea* genomic regions into *B. napus*. Actually, it is hard to draw a conclusion on the origin of the A and C subgenomes of *B. napus* only based on phylogenetic and population structure

analyses. Hence, we performed demographic analyses, and compared different alternative evolutionary models using $\partial a \partial i$, fastsimcoal2 and SMC++. To eliminate the impact derived from *B. rapa*, only winter landraces from Europe ($n = 10$) were used in our demographic model comparisons. Demographic modelling, using $\partial a \partial i$ and fastsimcoal2, both supported the models that the *B. napus* A subgenome evolved from the ancestor of European turnip, and the log-likelihood values of the two optimal models were -11826 and -4194976 (model a in Supplementary Fig. 2, model e in Supplementary Fig. 3). The best model in fastsimcoal2 also suggested that a gene flow event from European turnip to the *B. napus* A subgenome occurred ~ 106 – $1,170$ years ago (Supplementary Fig. 3). The demographic analysis results were also in accordance with previously reported archaeological or linguistic evidence, suggesting that our inference about the progenitor of *B. napus*, which was not affected by introgression events into *B. napus*, was reliable.

For ecotype improvement, we performed selection analyses using four different methods (F_{ST} , ROD, XP-CLR, and XP-EHH). The selection outlier regions were obtained when they were simultaneously detected by at least three of the four methods. In the revised manuscript, we indicate that several ecotype improvement-selection signals could be detected repeatedly using different methods (Supplementary Figs. 29 and 30). Comparisons between *B. napus* winter and semi-winter ecotypes, and between winter and spring ecotypes detected 1,996 and 1,117 overlapping outlier windows, including 4,548 genes in 156 selection regions and 2,729 genes in 107 selection regions, respectively (Supplementary Tables 32–35). The majority of selection regions were located in the C subgenome, and only 32 and 21 were distributed on the A subgenome, respectively. This suggested that ecotype improvement from winter to spring and semi-winter was caused by asymmetrical subgenomic selection. Furthermore, 844 outlier windows corresponding to 72 selection regions overlapped between two ecotype improvement analyses, and these parallel selection signals might contribute to the local adaptation of *B. napus*.

In the domestication analyses, we have tried to reduce the impact derived from *B. rapa*. Based on our aforementioned results, no significant effect from introgression

was found in our domestication analyses. For the improvement analyses, several overlapping improvement selection signals between winter and spring ecotypes, and between winter and semi-winter ecotypes also suggested that these selection signals were not affected by introgression during *B. napus* breeding.

2. There is also no mention of the fact there is preferential replacement of C genome regions with the A genome in *B. napus* through homoeologous recombination events (HE) (Chalhub et al, 2014), which is probably reflected in some of the differences that are observed between the A and C genomes. Interestingly the authors suggest that they looked for HE in their data (based on M&M) but do not refer to this analysis in the main text. Although their depth of sequencing (average ~5x) is probably too low to identify such events, so perhaps this should be removed from the methods, but the known prevalence of such events should be mentioned in the context of genome adaptation. For example, such events have been shown to impact flowering time and seed glucosinolate content.

RESPONSE:

Thank you for this comment. Homoeologous recombination events (HE) were important in *B. napus*, but the depth of sequencing (average ~5x) was too low to accurately identify such events in our study. Hence, methods and results concerning identification of HE have been removed in the revised manuscript.

In the GWAS results, we identified two significant loci associated with flowering time on the *B. napus* chromosomes A02 and C02, which contain two well-characterized flowering time genes *BnFLC.A2* and *BnFLC.C2*, respectively. A recent study reported that different *FLC* paralogs contributed differentially to natural variation in flowering time of *B. napus*, and that a 2.833-kb insertion in *BnFLC.A2* and its homeologous exchange (HE) with *BnFLC.C2* generated early-flowering *B. napus*. Since the genomic regions covering *BnFLC.C2* were substituted by homeologous A02 fragments via HE in several *B. napus* accessions, only partial accessions without HE could be effectively used to improve selection signal detection at the *BnFLC.C2* locus, suggesting that the accuracy of improvement selection

analyses could be increased if the HE events could be considered in the future. We have discussed this phenomenon in the revised manuscript.

3. With respect to the analyses completed, some elements do need to be clarified. It is not clear that the SNP numbers that are presented (eg. 733,165 BraA and 1,095,281 BolC) represent SNPs that would be informative across the whole dataset. It is certain that there would have been SNPs that were only informative among the *B. rapa* and the *B. napus* genotypes, respectively (similarly for the C genome). But if the authors only used a cut-off criteria of an allele frequency >1% or >0.03 (both values are indicated in M&M) across the whole population, which presumably means both *B. rapa* and *B. napus* in the case of the A genome, then SNPs that were specific to *B. rapa* and/or *B. napus* would be retained.

RESPONSE:

To clarify the SNP discovery process, we have revised the corresponding sentences as follows:

“We aligned the *B. napus* data to a *B. napus* ancestral pseudo-genome (merging the *B. rapa* and *B. oleracea* reference genomes; Methods) (Supplementary Fig. 1) and divided the SNPs into BnaA and BnaC two sets, based on the two progenitors. *B. rapa* and *B. oleracea* sequencing data were mapped onto the corresponding reference genomes. Then, the SNPs called from *B. rapa* were combined with BnaA to form the A (*B. rapa* and *B. napus*) subgenome SNP set (denoted as BraA, including 529,771 SNPs). Similarly, the C (*B. oleracea* and *B. napus*) subgenome SNP set (denoted as BolC), including 675,457 SNPs, was obtained (Supplementary Fig. 1).”

To increase the accuracy of our results, we obtained all the SNPs with a MAF of 0.05 and performed most of the analyses using three new SNP sets. Then, the A subgenome-specific SNPs derived from *B. napus* and *B. rapa* were used for population structure, LD, demographic, and domestication selection analyses (similarly to the C subgenome-specific SNPs). All the results have been updated in the revised manuscript.

4. The lengthy discussion of potentially important candidate genes that have contributed to the adaptation of *Brassica napus* has not really identified many novel candidates, most have been suggested in previous publications; however, the authors did present corroborating GWAS data, which certainly strengthened this section compared to some previous work.

RESPONSE:

We appreciate this comment. We have improved the selection analyses using four different methods (F_{ST} , ROD, XP-CLR, and XP-EHH). The selection outlier regions were obtained when they were simultaneously detected by at least three of the four aforementioned methods. Furthermore, we conducted GWAS using multi-locus random mixed linear models, increasing the accuracy of the GWAS results. Then, we combined the GWAS, improvement selection, and transcriptome analysis results to identify candidate genes regulating the 11 important traits in *B. napus*. We not only identified well-characterized key genes controlling the total glucosinolate content, erucic acid content, flowering time, and silique length (*GTR2*, *FAE1*, *FLC*, and *ARF18*), but also found several novel GWAS and improvement selection signals that overlapped. Candidate genes in these reliable QTL have been predicted, and merit further functional characterization.

Minor points:

5. P9, there were probably insufficient fodder and vegetable types represented in the dataset to make any comments on differential LD in these lines.

RESPONSE:

Thanks for the comment. *B. napus* is mainly used as an oilseed crop. Hence, it is hard to collect more fodder and vegetable type of *B. napus*, which may lead to inaccurate comparisons of LD decay among *B. napus* with different usages. These results have been moved to Supplementary Figs. 7–9.

6. P10, suggesting the C genome has lost diversity in comparison to the A genome, contradicts all evidence that the A genome diversity has been increased by multiple introgressions.

RESPONSE:

We recalculated the nucleotide diversity (π) for the European *B. napus* winter landraces and its two pseudo-ancestral subgenomes using new SNPs with MAF > 0.05. The results showed that “Nucleotide diversity (π) decreased from 7.23×10^{-4} in AA to 5.40×10^{-4} in AL and from 7.45×10^{-4} in CA to 4.97×10^{-4} in CL (Supplementary Tables 10, 11), implying that during domestication more genetic diversity was lost in the *B. napus* C subgenome than in the A subgenome.” Our demographic results suggested that introgression events from *B. rapa* and *B. oleracea* into *B. napus* both occurred ~1000 years ago, explaining why the nucleotide diversity of the tA genome is slightly higher than that of the C genome.

7. P17, the references used, no. 25 and 26, do not appear to support the statements.

RESPONSE:

We have checked all references in the manuscript to make sure that the references support the statements.

8. P6 and P25, there is a contradiction between the main text and the M&M, the main text indicates 103 SNPs were validated and the M&M only 20.

RESPONSE:

A total of 14 primer pairs (each containing 3–11 SNPs; Supplementary Table 5) for 103 SNPs were used to validate the accuracy. We have revised the main text and Methods accordingly. "A total of 103 SNPs were randomly selected for accuracy validation of SNP calling using conventional PCR and Sanger-based sequencing."

9. P28, “To mitigate the effect of LD, one SNP per 10 Kb was selected.” – Based on all previous publications and the authors own data this distance would be grossly insufficient to mitigate the effect of LD.

RESPONSE:

At a threshold of $r^2=0.3$, LD decay was 19.30 kb and 1365.30 kb in the A and C subgenomes of *B. napus*, respectively. There was a large difference between them. If we selected SNPs per 1000 kb, the number of SNPs for demographic history analyses would be less than 1000, and the results might be inaccurate. Therefore, we selected one SNP per 10 kb to ensure that the number of SNPs was sufficient to mitigate the effect of LD.

Reviewer #4 (Remarks to the Author):

In this manuscript, the authors describe the resequencing and analysis of 588 diverse *Brassica napus* (rapeseed) accessions aimed at investigating the parental origins, at the morphotype level, of this important allopolyploid crop. The origin of this crop remains poorly understood, in large part due to its polyploid nature and complex history of admixture during domestication. Furthermore, the progenitor species (*B. rapa* and *B. oleracea*) each have their own complex evolutionary histories riddled with polyploidy, hybridization and introgression events. The authors also identified genes associated with important agricultural traits using a combination of genomic and transcriptomic analyses. These findings and new resources have the potential to be a valuable resource to the community for future genomic studies and to develop tools to guide future breeding efforts.

However, after reviewing the methods, I have several major concerns:

1. A Neighbor-Joining (NJ) method was used to estimate relationships among resequenced accessions combined with publicly available data for progenitor species in an attempt to identify which morphotypes contributed to the origin of *B. napus*. This is a distance based approach, not a phylogenetic method, and typically results in estimates that are highly incongruent with a more rigorous likelihood methods for inferring phylogenies. Furthermore, the methods used by the authors lack any

statistical power. I strongly recommend that the authors reanalyze the data with a more rigorous phylogenetic method that uses an evolutionary model; maximum-likelihood approach (GTR+G Model) with bootstrapping to estimate node support. In short, the results from the current “phylogenetic” analyses are questionable and impact downstream analyses.

RESPONSE:

We appreciate this comment. To generate reliable phylogenetic trees, we firstly screened 17,000, 19377, and 19548 SNPs at fourfold-degenerate sites (MAF > 5%) from the BraA, BolC, and Bna SNP sets, respectively. Then, three maximum likelihood (ML) trees were constructed using IQ-TREE v1.6.6, according to the best model determined by the Bayesian information criterion (BIC). The best-fit models GTR+F+ASC+R5 (BIC log-likelihood = -13491934), GTR+F+ASC+R7 (BIC log-likelihood = -11162152), and TVM+F+ASC+R8 (BIC log-likelihood = -4979073) were chosen to construct the BraA, BolC, and Bna ML trees, respectively. The reliability of the ML trees was estimated using the ultrafast bootstrap approach (UFboot) with 1,000 replicates. An online tool Interactive tree of life (iTOL) v3 (<https://itol.embl.de>) was then used to display the three consensus trees.

We investigated the phylogenetic relationships among *B. napus* and different subspecies of two progenitors in the newly constructed ML trees and found that the topologies of current ML bootstrap trees were consistent with NJ trees in the original manuscript. Most *B. napus* accessions were clustered together based on ecotype, whereas clustering of *B. rapa* and *B. oleracea* largely reflected subspecies relationships. In the BraA ML tree, the *B. napus* clade was closest to the *B. rapa* ssp. *rapa* (European turnip) group and far from *B. rapa* ssp. *rapa* (Asian turnip) and other subspecies. In the BolC ML tree, the *B. napus* accessions were closest to a *B. oleracea* branch comprising kohlrabi, broccoli, cauliflower, and Chinese kale. These phylogenies were in accordance with the demographic models of the *B. napus* evolutionary history analyzed by *∂a∂i* and fastsimcoal.

2. This study lacks sampling of any wild A subgenome subspecies e.g. field mustard (*B. rapa* subsp. *sylvestris*). Thus, the authors should not conclude that European turnip types are the A subgenome ancestor contributed to present day *B. napus*. I suggest that the authors either modify the text to reflect this taxon sampling problem or reanalyze the data with additional sampling of wild A subgenome species (See Guo et al. 2014 or Qi et al. 2016). It is possible that the appropriate lineages just weren't sampled. Also see below minor comment 4)

Guo, Y., Chen, S., Li, Z., & Cowling, W. A. (2014). Center of origin and centers of diversity in an ancient crop, *Brassica rapa* (turnip rape). *Journal of Heredity*, 105(4), 555-565.

Qi, X., An, H., Ragsdale, A. P., Hall, T. E., Gutenkunst, R. N., Chris Pires, J., & Barker, M. S. (2017). Genomic inferences of domestication events are corroborated by written records in *Brassica rapa*. *Molecular ecology*, 26(13), 3373-3388.

RESPONSE:

Guo et al. 2014 suggested that *B. rapa* var. *sylvestris* might be the wild-type *B. rapa* accessions. Using expanded samples of rapini (*B. rapa* subsp. *sylvestris*), brown sarson (*B. rapa* subsp. *dichotoma*), and yellow sarson (*B. rapa* subsp. *trilocularis*), a recent study found no evidence to support the contention that rapini is the wild type or the earliest domesticated subspecies of *B. rapa*. We have discussed our sample limitation in the revised manuscript as follows:

"Previous archaeological and linguistic lines of evidence suggest that turnip is likely the first domesticated *B. rapa* in the European-Central Asian region^{42, 43}. A recent demographic inference further supported an eastward series of *B. rapa* domestication events, over the past several thousand years, and rapini (*B. rapa* ssp. *sylvestris*), which split from the European-Central Asian *B. rapa* (European turnip) cluster, approximately 3715–6190 years ago, is not likely a wild *B. rapa*⁴⁴. Considering the origin time and location of *B. napus* in previous studies, European turnip might be the only possible A subgenome donor for *B. napus* formation. Sampling of more wild *B. rapa* and *B. oleracea* relatives would be helpful for better understanding the complex events that occurred during *B. napus* origin."

3. Lines 146-148. "..., our data support the model indicating that the ancestor of *B. napus* split from the common ancestor of four *B. oleracea* subspecies at about 7,500–12,500 years ago, ...". These date estimates predate the origin of all the *B. oleracea* morphotypes. The origin of *B. napus*, if it only involved these four subspecies (i.e. morphotypes), obviously could not have occurred prior to the domestication of *B. oleracea* (i.e. origin of all cruciferous vegetables). This is likely due to the methods used to estimate relationships (see #1 above). Also, strong bootstrap support values are needed to confidently identify progenitor species.

RESPONSE:

Thank you for raising this important concern. As in our abovementioned response, we have constructed the ML tree using SNPs at fourfold-degenerate sites in the BolC SNP sets, based on the best-fit model GTR+F+ASC+R7 (BIC log-likelihood = -11162152). The reliability of the ML trees was estimated using the ultrafast bootstrap approach (UFboot) with 1,000 replicates. In the BolC ML tree, the *B. napus* accessions were also closest to a *B. oleracea* branch comprising kohlrabi, broccoli, cauliflower, and Chinese kale with high bootstrap values, which could be observed in the revised Fig. 3a. Both the phylogenies and demographic models of the *B. napus* evolutionary history analyzed by $\partial\text{a}\partial\text{i}$ and fastsimcoal supported the conclusion that *B. napus* originated from the common ancestor of four *B. oleracea* subspecies.

As for the origin time of *B. napus* (about 7,500–12,500 years ago), it was estimated by Chalhoub et al. 2014 using the synonymous substitution of orthologous gene pairs between *B. rapa* ssp. *pekinensis* (Chiifu-401-42, Chinese cabbage) and the A subgenome of *B. napus*, and between *B. oleracea* var. *capitata* (line 02–12, cabbage) and the C subgenome of *B. napus*. However, the accuracy of the origin time of *B. napus* needs to be supported by more evidence, as the results might be affected by the progenitors used in the analyses. Hence, we have deleted the origin time of *B. napus* in the revised manuscript, though it still has to be used in our demographic analysis as a necessary parameter.

4. Lines 199-201. "... the C subgenome of *B. napus* might be caused by the multiplicity of origins of that subgenome." See below minor comment #1. I suspect that multiple origins of *B. napus* is far more likely than a single origin; given that hybridization between these progenitor species and polyploidization can spontaneously happens in gardens and greenhouses.

RESPONSE:

Thanks for the concern. Our data supported that the *B. napus* A subgenome evolved from the ancestor of European turnip with a gene flow event from European turnip into the *B. napus* A subgenome that occurred ~106–1,170 years ago (Supplementary Fig. 3), and the *B. napus* C subgenome originated from the common ancestor of kohlrabi, cauliflower, and broccoli with a recent gene flow into *B. napus* ~108–898 years ago. According to Qu et al. 2014, several *B. rapa* subspecies (such as *B. rapa* subsp. *pekinensis*, *B. rapa* subsp. *Chinensis*, and *B. rapa* subsp. *trilocularis*) may have evolved due to an eastward *B. rapa* domestication of European-Central Asian *B. rapa* (including European turnip). Hence, it is not surprising that some descendent *B. rapa* may have retained the ability to hybridize with *B. oleracea*, like their ancestor, European turnip. We believe these hybridizations may reflect the gene flow events from *B. rapa* and *B. oleracea* into *B. napus* revealed in our demographic analyses.

5. I would be hesitant to estimate copy number variation with only 3-7X sequence depth especially in a highly polyploid species. There is just too much variation in coverage at this sequence depth to accurately call presence-absence or even copy number variation. I would recommend removing this from your analyses. Have you experimentally validated any of these 3,639 CNVs (lines 88-89)?

RESPONSE:

Thank you for pointing out this problem. Due to the large number of *B. napus* accessions, we only obtained a read depth average of ~5× and range from 3.37× to 7.71×, which is not sufficient for CNV and HE identification in *B. napus* and may lead

to incorrect results. Hence, we have removed these parts from the Results and Methods in the revised manuscript.

Additional minor comments and suggestions:

6. Lines 62-65 “The precise identities of the two progenitors that hybridized to form *B. napus* remain elusive, as *B. rapa* and *B. oleracea* occur in geographically distinct areas and each has morphologically diverse subspecies.” This is not entirely correct. These species have been commonly cultivated together throughout Europe for hundreds of years. Hybridization between these species is common, and hybrids are known to spontaneously double to polyploids. I’ve personally observed this in the greenhouse. This is why present day *B. napus* diversity is unlikely a single origin.

RESPONSE:

Thank you for raising this important concern. We have realized that the original description in lines 62–65 is not completely correct. To better explain why the precise identities of the two progenitors that hybridized to form *B. napus* remain elusive, we have changed the sentence to “The precise identities of the two progenitors that hybridized to form *B. napus* remain elusive, as *B. rapa* and *B. oleracea* have morphologically diverse subspecies and have commonly been cultivated throughout Europe for hundreds of years. The natural hybridization among these species occurred occasionally under appropriate conditions”.

7. Why was flowering time further investigated given that it wasn’t highly enriched in the GO term enrichment analysis?

RESPONSE:

Flowering time is one of the most important traits, determining the cultivation areas of *B. napus*. Genes in the ecotype improvement selection regions were not over-enriched in GO terms related to flowering-time pathways, possibly due to the unstable improvement selection signals derived from the individual method.

To increase the accuracy of the results, we detected improvement-selection signals using another two methods XP-CLR (cross-population composite likelihood ratio test) and XP-EHH (cross-population extended haplotype homozygosity), in addition to F_{ST} and ROD. Then, the selection outlier regions were obtained when they were simultaneously detected by at least three of the four methods (F_{ST} , ROD, XP-CLR, and XP-EHH). We found that genes in the ecotype improvement selection regions were over-represented in maintenance of floral organ identity (GO:0048497), floral organ abscission (GO:0010227), and regulation of floral meristem growth (GO:0010080), suggesting that these improvement selection signals might be critical for local environmental adaptation of *B. napus* (Supplementary Tables 36, 37). We have corrected the text in the revised manuscript accordingly.

8. Lines 73-75: “In this study, we performed deep genome sequencing of 588 *B. napus* accessions and transcriptome sequencing of 11 tissues from two *B. napus* accessions with different seed quality.” 3-8x sequence depth is not “deep genome sequencing”. This would be considered “genome skimming” or “shallow genome sequencing”.

RESPONSE:

Thank you for the good suggestion. We have changed “deep genome sequencing” to “shallow genome sequencing” in the revised manuscript.

9. To support claims of a turnip (*B. rapa ssp rapa*) origin for the A subgenome, further evidence should be provided that domesticated turnips existed during time of *B. napus* formation (7,500-12,000 years ago). Highlighting archaeological or linguistic evidence would greatly strengthen this argument (e.g. Ignatov et al. 2008 and Reiner 1995)

Ignatov, A. N., Artemyeva, A. M., & Hida, K. (2008, September). Origin and expansion of cultivated *Brassica rapa* in Eurasia: linguistic facts. In V International Symposium on Brassicas and XVI International Crucifer Genetics Workshop, Brassica 2008 867 (pp. 81-88).

Reiner, H., Holzner, W., & Ebermann, R. (1995). The development of turnip-type and oilseed-type *Brassica rapa* crops from the wild-type in Europe—An overview of botanical, historical and linguistic facts. *Rapeseed Today Tomorrow*, 4, 1066-1069.

RESPONSE:

Thanks for your valuable suggestion. We have added archaeological evidence and demographic analyses using *ada* and *fastsimcoal* to support our claims of a turnip (*B. rapa* ssp. *rapa*) origin for the A subgenome of *B. napus*. The revised paragraph is as follows:

“Previous archaeological and linguistic lines of evidence suggest that turnip is likely the first domesticated *B. rapa* in the European-Central Asian region^{42, 43}. A recent demographic inference further supported an eastward series of *B. rapa* domestication events, over the past several thousand years, and rapini (*B. rapa* ssp. *sylvestris*), which split from the European-Central Asian *B. rapa* (European turnip) cluster, approximately 3715 – 6190 years ago, is not likely a wild *B. rapa*⁴⁴. Considering the origin time and location of *B. napus* in previous studies, European turnip might be the only possible A subgenome donor for *B. napus* formation.”

Reviewers' comments:

Reviewer #1 (Remarks to the Author):

Many thanks for the opportunity to read the revised manuscript. I believe it is much improved, however I still have issue with the term 'domestication' in the context of *B. napus*. There are no wild *B. napus*. It was formed from the hybridisation of domesticated diploid progenitors, so was not in itself domesticated. There has certainly been selection in *B. napus*, especially regarding glucosinolates and erucic acid, and this is well documented, but any reference to domestication should be made in relation to the diploid progenitors and not the polyploid.

Reviewer #2 (Remarks to the Author):

the imputation accuracy (correlation of real vs. imputed genotypes) across a range of MAFs within the subpopulations needs to be calculated within BEAGLE. Imputation accuracy based on 19 SNPs is not valuable.

Reviewer #3 (Remarks to the Author):

The authors have made substantial changes to the manuscript; including re-analyses and additional analyses, and have removed some of the more contentious elements. I have no further suggestions for improvement.

Reviewer #4 (Remarks to the Author):

The authors largely addressed my primary concerns in the revised version of the manuscript. However, I would strongly encourage the authors to add new text to both the Introduction and Discussion that outlines their proposed model, compared to other possible models, for the origin and domestication of *Brassica napus*. This includes models that wild species vs cultivars were parental progenitors. It's important to put your model in context with previous analyses and date estimates (see my previous comment #3). I would highly recommend including a timeline in the Discussion for the origin, domestication and modern crop improvement efforts of *B. napus*.

Instead of deleting the "origin time of *B. napus* (about 7,500–12,500 years ago)", this should be discussed as a central finding in this manuscript. Highlighting this major finding is important for future discussion.

"However, the accuracy of the origin time of *B. napus* needs to be supported by more evidence, as the results might be affected by the progenitors used in the analyses." There are multiple genomes available now for *B. rapa* and *B. oleracea* -- thus could be tested by the authors.

Responses to reviewers' comments

Reviewers' comments:

Reviewer #1 (Remarks to the Author):

Many thanks for the opportunity to read the revised manuscript. I believe it is much improved, however I still have issue with the term 'domestication' in the context of *B. napus*. There are no wild *B. napus*. It was formed from the hybridisation of domesticated diploid progenitors, so was not in itself domesticated. There has certainly been selection in *B. napus*, especially regarding glucosinolates and erucic acid, and this is well documented, but any reference to domestication should be made in relation to the diploid progenitors and not the polyploid.

RESPONSE:

Thank you for raising this comment. We have accepted the reviewer's suggestion and deleted the domestication in the revised manuscript. In the revised manuscript, we divided the *B. napus* evolution process into two improvement stages after origin of *B. napus*. The first stage of improvement (FSI) was the process from pseudo-wild ancestral subgenomes of original *B. napus* to *B. napus* landrace, while the second stage of improvement (SSI) represent the process from landraces to improved cultivars. All the descriptions of domestication and improvement in the original manuscript have been updated now based on abovementioned principles.

Besides, we maintained the comparisons for seed quality improvement and ecotype improvement and didn't make change to these contents.

Reviewer #2 (Remarks to the Author):

the imputation accuracy (correlation of real vs. imputed genotypes) across a range of MAFs within the subpopulations needs to be calculated within BEAGLE. Imputation accuracy based on 19 SNPs is not valuable.

RESPONSE:

Thanks for your valuable suggestion. In the revised manuscript, we estimated the imputation accuracy using two measures. Based on the results, we generated a new Supplementary Fig. 16 to display the results of imputation accuracy evaluation derived from two biological replicates of 20 *B. napus* accessions in intervals of 5% of the MAF. Then, we added the descriptions as following “One is the comparison of the imputation results of 19 polymorphic nucleotides with the corresponding Sanger sequencing results (Supplementary Table 34). The other was the correlations (r^2) between imputed and true genotypes, which were calculated at each locus for 20 accessions of biological replicates (R021 ~ R040, Supplementary Table 1) in intervals of 5% of MAF. Missing SNPs in the true genotypes were excluded when calculating the correlations.” and “The average correlations (r^2) between imputed and true genotypes for two biological replicates of 20 accessions was 0.956 with minimum and maximum values ranging from 0.928 to 0.967 (Supplementary Fig. 16), further confirming the accuracy of the imputed genotypes.” to the Methods and Results sections in the revised manuscript, respectively.

Reviewer #3 (Remarks to the Author):

The authors have made substantial changes to the manuscript; including re-analyses and additional analyses, and have removed some of the more contentious elements. I have no further suggestions for improvement.

RESPONSE:

Thank you for your comment.

Reviewer #4 (Remarks to the Author):

The authors largely addressed my primary concerns in the revised version of the manuscript. However, I would strongly encourage the authors to add new text to both the Introduction and Discussion that outlines their proposed model, compared to other possible models, for the origin and domestication of *Brassica napus*. This includes models that wild species vs cultivars were parental progenitors. It's important to put your model in context with previous analyses and date estimates (see my previous comment #3). I would highly recommend including a timeline in the Discussion for the origin, domestication and modern crop improvement efforts of *B. napus*.

Instead of deleting the "origin time of *B. napus* (about 7,500–12,500 years ago)", this should be discussed as a central finding in this manuscript. Highlighting this major finding is important for future discussion.

"However, the accuracy of the origin time of *B. napus* needs to be supported by more evidence, as the results might be affected by the progenitors used in the analyses." There are multiple genomes available now for *B. rapa* and *B. oleracea* -- thus could be tested by the authors.

RESPONSE:

Thanks for your valuable suggestions. We have re-organized the background introduction and added more sentences and previous reports to describe the proposed model for origin and evolutionary history of *B. napus* in the revised Introduction.

The following sentences "As one of the earliest allopolyploid crops, *B. napus* was formed by hybridization of *B. rapa* and *B. oleracea*¹. The estimated formation time of *B. napus* were ~6,700⁴ and ~7,500 years ago⁵ and 0.038–0.051 million years ago⁶, based on two synonymous substitution (Ks) estimations and a Bayesian Markov chain Monte Carlo (MCMC) simulation, respectively. The literatures recorded that winter *B. napus* was first cultivated in Europe⁷. Around the year 1,700, spring *B. napus* was developed and spread to England in the late 18th century⁸. The semi-winter ecotype was mainly cultivated in China, which was introduced from Europe in the 1,930–1,940s⁹." and "Although a recent study suggested that the *B. napus* A

subgenome might be derived from the ancestor of European turnip (*B. rapa* ssp. *rapa*)⁶, more evidence to support the conclusion need to be provided, due to only 5 *B. napus* and 27 *B. rapa* accessions were included in their analysis. Previous studies based on nuclear and chloroplast markers also suggested that *B. napus* may have developed from an interspecific cross between turnip and broccoli, or resulted from several independent hybridization events^{10,11}. To further understand the evolution of *B. napus*, it is necessary to reveal whether wild species or domesticated donors were parental progenitors, which *B. rapa* and *B. oleracea* subspecies involved in the formation of *B. napus*.” have been added to the Introduction section.

To better explain the different evolutionary events using a timeline method in the Discussion section, we also generated the Supplementary Fig. 32 to illustrate the proposed model for origin and evolutionary history of *B. napus*. Then, we made further discussion with previous researches and historical records to summarize the origin time of *B. napus*, the original form of *B. napus*, and different divergence events in different ecotype and usage of *B. napus*. We think that these descriptions and corresponding new figure provide further insight into the evolution of *B. napus*. In the revised Discussion, we have revised the first paragraph as follows:

“In this study, we developed a large genome variation data set for genetically diverse *B. napus* accessions, which provided an opportunity to finely resolve the origin and evolutionary history of *B. napus*. Based on aforementioned analyses, we posit that the *B. napus* was originated from the hybridization between domesticated *B. rapa* and *B. oleracea* ~1.91 – 7.18 thousand years ago (Supplementary Fig. 32), which accorded with previous conclusions (~6,700 and 7,500 years ago) derived from Ks estimation^{4,5}. The *B. napus* A subgenome evolved from the ancestor of European turnip; and the *B. napus* C subgenome might have evolved from the common ancestor of kohlrabi, cauliflower, broccoli, and Chinese kale. In addition, the LD and demographic analyses support that the original *B. napus* was winter oilseed, and the spring and semi-winter *B. napus* developed ~416 and ~60 years ago, and non-oilseed *B. napus* developed ~277 years ago (Supplementary Fig. 32). These results are consistent with historical records, which indicate that spring *B. napus* developed

around the year 1,700⁸, and that the semi-winter ecotype has only a short history in China, and arose from the winter ecotype, which was introduced from Europe in the 1,930–1,940s⁹. In recent 1,000 years, gene flow from two progenitors into *B. napus* also occurred occasionally, leading to improvement of complex traits in *B. napus*.”

REVIEWERS' COMMENTS:

Reviewer #2 states in the Remarks to Editors sections that (s)he has not further comment.

Reviewer #4 (Remarks to the Author):

The authors have adequately addressed all of my major concerns.

A couple of very minor edits -- The commas should be removed from dates in the new text. For example, "1,930-1,940s" should read "1930-1940s". Also "... around the year 1,700" should be "... around the year 1700". Lastly, lines 51-52 -- please use consistent units for dates (years or millions years ago).

These are all minor edits (there are many others) that I assume will be corrected later by the copy editor.

REVIEWERS' COMMENTS:

Reviewer #2 states in the Remarks to Editors sections that (s)he has not further comment.

Response: Thank you for your comment.

Reviewer #4 (Remarks to the Author):

The authors have adequately addressed all of my major concerns.

A couple of very minor edits -- The commas should be removed from dates in the new text. For example, "1,930-1,940s" should read "1930-1940s". Also "... around the year 1,700" should be "... around the year 1700". Lastly, lines 51-52 -- please use consistent units for dates (years or millions years ago).

These are all minor edits (there are many others) that I assume will be corrected later by the copy editor.

Response: The commas from dates have been removed, and the units for dates were also been unified in the revised manuscript.